# Heterosynaptic plasticity of the visuo-auditory projection requires cholecystokinin released from entorhinal cortex afferents

**Wenjian Sun[1,2†‡], Haohao Wu[3†§], Yujie Peng[1,2†#], Xuejiao Zheng[1,2†¶], Jing Li[1,2†], Dingxuan Zeng[1†], Peng Tang[1,2], Ming Zhao[3], Hemin Feng[1,2**], Hao Li[1,2], Ye Liang[1,2], Junfeng Su[1††], Xi Chen[1,4*], Tomas Hökfelt[3,5*], Jufang He[1,4*]**

[1]Department of Neuroscience, City University of Hong Kong, Hong Kong, China; [2]Centre for Regenerative Medicine and Health, Hong Kong Institute of Science & Innovation, Chinese Academy of Sciences, Hong Kong, China; [3]Department of Neuroscience, Karolinska Institutet, Stockholm, Sweden; [4]City University of Hong Kong Shenzhen Research Institute, Shenzhen, China; [5]Institute of Advanced Study, City University of Hong Kong, Hong Kong, China

**\*For correspondence:**
xi.chen@cityu.edu.hk (XC);
Tomas.Hokfelt@ki.se (TH);
jufanghe@cityu.edu.hk (JH)

[†]These authors contributed equally to this work

**Present address:** [‡]Zilkha Neurogenetic Institute, University of Southern California, Los Angeles, United States; [§]Biozentrum, Department of Cell Biology, University of Basel, Basel, Switzerland; [#]Friedrich Miescher Institute for Biomedical Research, Basel, Switzerland; [¶]Beijing Genomics Institute-Shenzhen, Shenzhen, China; [**]Department of Neurosurgery, Stanford University School of Medicine, Stanford, United States; [††]F.M. Kirby Neurobiology Center, Boston Children's Hospital, Boston, United States

**Competing interest:** The authors declare that no competing interests exist.

**Abstract** The entorhinal cortex is involved in establishing enduring visuo-auditory associative memory in the neocortex. Here we explored the mechanisms underlying this synaptic plasticity related to projections from the visual and entorhinal cortices to the auditory cortex in mice using optogenetics of dual pathways. High-frequency laser stimulation (HFS laser) of the visuo-auditory projection did not induce long-term potentiation. However, after pairing with sound stimulus, the visuo-auditory inputs were potentiated following either infusion of cholecystokinin (CCK) or HFS laser of the entorhino-auditory CCK-expressing projection. Combining retrograde tracing and RNAscope in situ hybridization, we show that *Cck* expression is higher in entorhinal cortex neurons projecting to the auditory cortex than in those originating from the visual cortex. In the presence of CCK, potentiation in the neocortex occurred when the presynaptic input arrived 200 ms before postsynaptic firing, even after just five trials of pairing. Behaviorally, inactivation of the CCK[+] projection from the entorhinal cortex to the auditory cortex blocked the formation of visuo-auditory associative memory. Our results indicate that neocortical visuo-auditory association is formed through heterosynaptic plasticity, which depends on release of CCK in the neocortex mostly from entorhinal afferents.

## Editor's evaluation

This fundamental work advances our understanding of how neuropeptides influence cortical circuits and cortical plasticity. The evidence supporting the conclusions is compelling. This work would be of interest to neuroscientists working on cortical processing and plasticity and general roles neuropeptides play in brain function.

## Introduction

Cross-modal association is crucial for our brain to integrate information from different modalities. This integrated processing is fundamental for creating complete and context-rich memories. Traditionally, it is assumed to mainly occur in higher-order association cortices as evidenced by both anatomical

(*Cusick et al., 1995*; *Seltzer et al., 1996*) and physiological (*Fuster et al., 2000*; *Lipton et al., 1999*; *Sakai and Miyashita, 1991*; *Schlack et al., 2005*; *Sugihara et al., 2006*) studies. Evidence from fMRI studies (*Calvert et al., 1997*; *Finney et al., 2001*; *Foxe et al., 2002*; *Pekkola et al., 2006*) and in vivo electrophysiological recordings (*Brosch et al., 2005*; *Zhou and Fuster, 2004*; *Zhou and Fuster, 2000*) has also indicated the involvement of unimodal sensory cortices. With respect to the visuo-auditory association, we have demonstrated that neurons in the auditory cortex (AC) start to respond to light stimuli after a classical fear conditioning, coupling light and electrical stimulation (ES) of the AC, and this type of association can be blocked by inactivation of the entorhinal cortex (EC) (*Chen et al., 2013*). This is consistent with that removal of the bilateral medial temporal lobe, where the EC locates, prevents the formation of long-term declarative memory in patient H.M. (*Milner and Klein, 2016*; *Scoville and Milner, 1957*).

The EC is strongly and reciprocally connected with both the hippocampus and neocortex (*Canto et al., 2008*; *Swanson and Köhler, 1986*), and rich in cholecystokinin (CCK)-positive neurons, both in rat (*Greenwood et al., 1981*; *Innis et al., 1979*; *Köhler and Chan-Palay, 1982*) and mouse (*Meziane et al., 1997*). As the most abundant neuropeptide in brain (*Beinfeld et al., 1981*; *Dockray et al., 1978*; *Innis et al., 1979*; *Larsson and Rehfeld, 1979*; *Rehfeld, 1978*; *Vanderhaeghen et al., 1980*), sulphated cholecystokinin octapeptide (CCK-8S) was shown to have excitatory effects on pyramidal neurons (*Dodd and Kelly, 1981*) and plays an important role in learning and memory (*Horinouchi et al., 2004*; *Lo et al., 2008*; *Meziane et al., 1993*; *Nomoto et al., 1999*; *Tsutsumi et al., 1999*). The neocortex expresses a variety of neuropeptides, primarily in GABAergic, inhibitory interneurons (*Hendry et al., 1984*; *Somogyi et al., 1984*; *Somogyi and Klausberger, 2005*). CCK is, however, also found in pyramidal projection neurons, which have high levels of *Cck* transcript as shown with in situ hybridization (*Burgunder and Young, 1988*; *Schiffmann and Vanderhaeghen, 1991*; *Siegel and Young, 1985*). In line with these findings, we have previously shown that the cortical projecting neurons in the EC of mouse and rat mostly are CCK[+] and glutamatergic, and are important for visuo-auditory association (*Chen et al., 2019*; *Li et al., 2014*; *Zhang et al., 2020*). However, how various pathways are involved and the underlying mechanism to establish the visuo-auditory association is still not clear.

In the present study, we expanded our research on visuo-auditory association by using two channelrhodopsins, Chronos and ChrimsonR, to examine various ways to potentiate the visual cortex (VC) to AC projection: (i) high-frequency laser stimulation (HFS laser) of VC-to-AC projection with classical high-frequency stimulation protocol; (ii) infusion of a CCK agonist in the AC followed by (a) pairing of presynaptic activation of VC-to-AC terminals expressing opsin by single pulse laser stimulation with postsynaptic noise-induced AC firing (CCK + Pre/Post Pairing), (b) presynaptic activation of VC-to-AC terminals (CCK + Pre), (c) postsynaptic noise-induced AC firing (CCK + Post), and (d) nothing (CCK alone); (iii) HFS laser of EC-to-AC CCK[+] projection followed by (a) pairing of presynaptic activation of VC-to-AC terminals expressing opsin by single pulse laser stimulation with postsynaptic noise-induced AC firing (HFS laser EC-to-AC + Pre/Post Pairing), (b) presynaptic activation of VC-to-AC terminals (HFS laser EC-to-AC + Pre), (c) postsynaptic noise-induced AC firing (HFS laser EC-to-AC + Post), (d) nothing (HFS laser EC-to-AC alone); (iv) HFS laser of VC-to-AC projection followed by Pre/Post Pairing (HFS laser VC-to-AC + Pre/Post Pairing); and (v) testing different parameters of the pairing protocol: the frequency of the laser stimulation of the EC-to-AC CCK[+] projection; the delay between the termination of HFS laser of EC-to-AC CCK[+] projection and Pre/Post Pairing (Delay 1); and the delay between presynaptic activation of VC-to-AC projection and postsynaptic AC activation (Delay 2). Of particular interest was to test spike timing-dependent plasticity (STDP), an extension of the Hebbian learning rule, stating that, in order to induce potentiation, the critical window between the arrival of presynaptic input and postsynaptic firing should not be more than 20 ms (*Bi and Poo, 1998*; *Markram et al., 1997*; *Zhang et al., 1998*). This theory has been challenged (*Bittner et al., 2017*; *Drew and Abbott, 2006*; *Izhikevich, 2007*), and we hypothesized that endogenous CCK could be involved in a type of synaptic potentiation that is different from STDP. Furthermore, behavioral experiments were conducted to assess the impact of disrupting the EC-to-AC CCK[+] projection or the VC-to-AC projection on the establishment and recall of visuo-auditory associations. Besides, we used retrograde tracing and RNAscope in situ hybridization to analyze *Cck* expression in EC neurons projecting to the AC versus those from the VC and examined whether CCK was released following HFS of the EC-to-AC CCK[+] projection by using CCK sensor.

## Results

### The auditory cortex receives a direct projection from the visual cortex

To examine the origin of the visual information underlying the previously observed visual responses in the AC (*Chen et al., 2013*; *Li et al., 2014*), we injected the retrograde tracer cholera toxin subunit B (Alexa Fluor 488 Conjugate) in the AC. The auditory thalamus was strongly labeled in the dorsal (MGD), ventral (MGV), and medial (MGM) subdivisions (*Figures 1A1 and 2*). Many retrogradely labeled neurons were observed in both the primary and associative VC (*Figure 1A3 and 4*), but more were observed in the associative than the primary VC and were mainly distributed in the layer V (*Figure 1*, *Figure 1—figure supplement 1A*, two-way ANOVA, $F_{(4, 40)}$ = 4.707, significant interaction, p=0.0033, n = 5; $F_{(1,40)}$ = 9.768, primary [6.3 ± 1.0] vs. associative [11.1 ± 2.2], p=0.0033, 95% confidence interval [CI] of difference [–7.8 to –1.7]; $F_{(4, 40)}$ = 17.18, different layers, p<0.0001; Bonferroni's multiple comparison test, Layer I-Associative VC [0.7 ± 0.4] vs. Layer V-Associative VC [28.9 ± 4.7], p<0.0001, 95% CI of difference [–40.2 to –16.2]; Layer II/III-Associative VC [6.3 ± 1.7] vs. Layer V-Associative VC [28.9 ± 4.7], p<0.0001, 95% CI of difference [–34.5 to –10.6]; Layer IV-Associative VC [10.7 ± 2.9] vs. Layer V-Associative VC [28.9 ± 4.7], p=0.0002, 95% CI of difference [–30.2 to –6.3]; Layer VI-Associative VC [8.8 ± 2.1] vs. Layer V-Associative VC [28.9 ± 4.7], p<0.0001, 95% CI of difference [–32.1 to –8.2], see *Supplementary file 1* for detailed statistics). The result here is consistent with previous studies reporting the existence of reciprocal projections between the VC and AC (*Bizley et al., 2007*; *Budinger et al., 2006*; *Falchier et al., 2002*; *Falchier et al., 2010*; *Rockland and Ojima, 2003*), and provides a possible anatomical basis for the visual inputs of visuo-auditory associations formed in the AC.

### HFS of the VC-to-AC projection does not induce long-term potentiation (LTP)

HFS is a classical protocol to induce LTP (*Bashir et al., 1991*; *Bliss and Gardner-Medwin, 1973*; *Bliss and Lømo, 1973*; *Hernandez et al., 2005*; *Yun et al., 2002*), which typically consists of 1 s train of pulses at 100 Hz repeated three times with an inter-trial interval (ITI) of 10 s. Based on the current understanding, we should be able to induce LTP if the VC-to-AC projection is activated with HFS laser. We injected AAV9-syn-ChrimsonR-tdtomato in the VC of wildtype mice and manipulated the VC projection terminals in the AC expressing ChrimsonR (*Figure 1B*), a variant of channelrhodopsin-2. Laser stimulation of the VC-to-AC projection induced a field excitatory postsynaptic potential (fEPSP$_{VC-to-AC}$) in the AC as an indicator of the VC-to-AC input. To prevent photoelectric artifacts, fEPSPs evoked by laser stimulation were recorded by glass pipette electrodes with an impedance of 1 MΩ rather than metal electrodes (*Cardin et al., 2010*; *Kozai and Vazquez, 2015*; *Figure 1—figure supplement 1B and C*). The recording of fEPSPs was conducted across layers II/III to layer V within the AC (*Figure 1B*). Generally, a laser with higher intensity induced an fEPSP with a steeper slope and larger amplitude until saturation was reached (*Figure 1C*). Considering the kinetics of ChrimsonR (*Klapoetke et al., 2014*), we modified the HFS protocol and used four trials of 1 s pulse train at 80 Hz with an ITI of 10 s (*Figure 1D*, upper, *Figure 1—figure supplement 1D*). We chose the laser intensity that induced a 50% fEPSP saturation for baseline and the post-HFS tests and 75% for HFS laser. However, no significant LTP was induced in the VC-to-AC projection by HFS laser of this pathway alone (*Figure 1D*, bottom, paired *t*-test, t(9) = 0.878, before [101.7 ± 1.8%] vs. after [95.6 ± 7.1%], 95% CI of difference [–9.5% to 21.6%], p=0.403, n = 10, see *Supplementary file 1* for detailed statistics).

### VC-to-AC inputs are potentiated after pairing the activation of their terminals with noise bursts in the presence of CCK

Hebbian theory claims that cells that fire together wire together. We next tested whether VC-to-AC inputs can be potentiated after pairing with repetitive AC activation. We used laser stimulation of VC-to-AC projection to evoke presynaptic input and noise stimulus to trigger postsynaptic AC firings. Since the latency of fEPSP$_{VC-to-AC}$ is approximately 2–2.5 ms, and the firing latency of noise responses in the AC of mice is mostly equal to or longer than 13 ms, we presented the laser stimulus 10 ms after noise. Therefore, we activated the presynaptic input just before the postsynaptic firing (Pre/Post Pairing, *Figure 1E*). Responses to noise at different sound intensities were first tested (*Figure 1F*), and

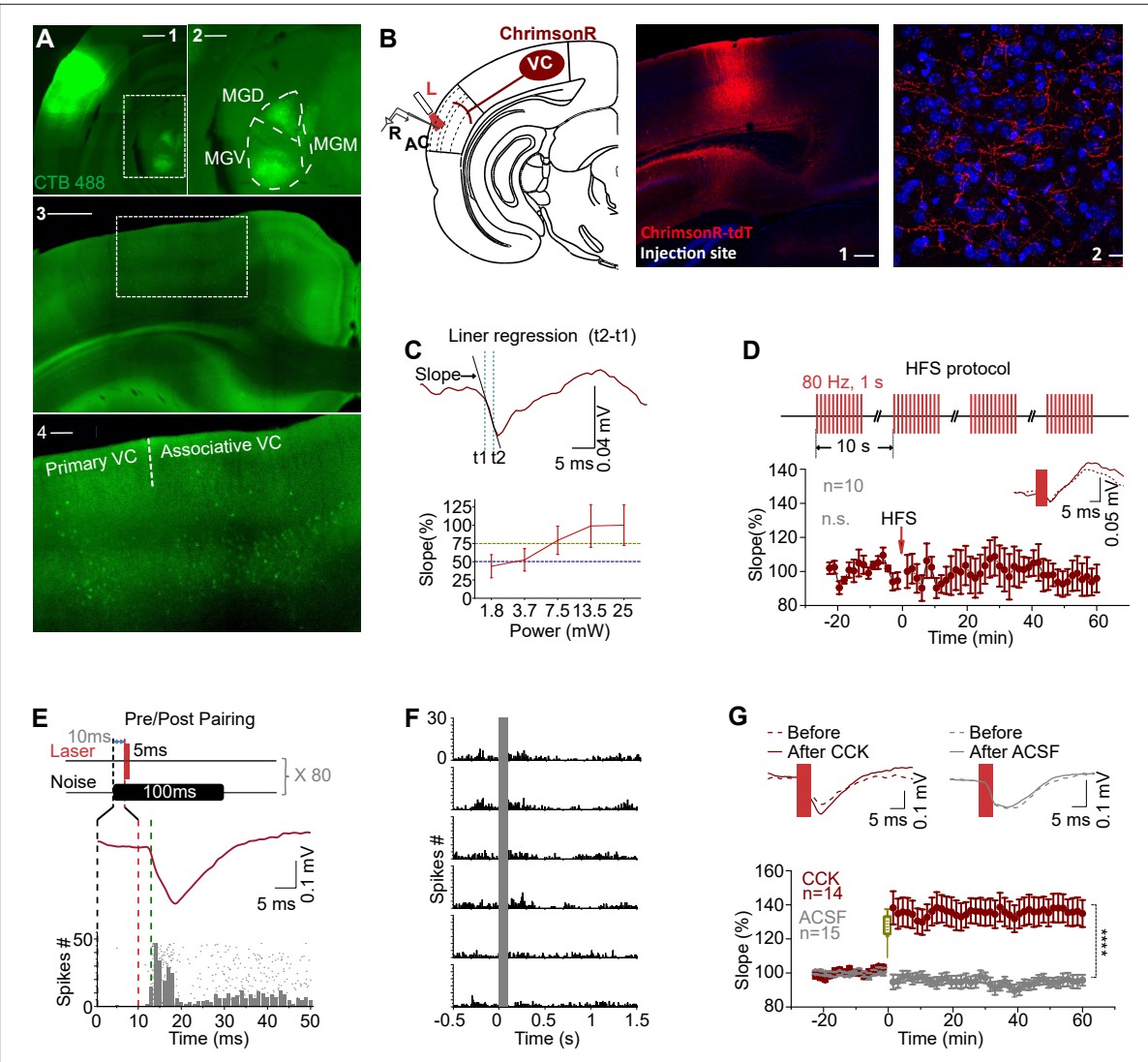

**Figure 1.** Visual cortex-to-auditory cortex (VC-to-AC) projection was potentiated after pairing laser stimulation evoked presynaptic activation with noise-induced postsynaptic firing in the presence of CCK. (**A**) Images show the injection site of the retrograde tracer (CTB488) in the AC (**A1**, scale bar: 1000 μm) and retrogradely labeled neurons in the auditory thalamus (**A2**, an enlargement of the boxed area in **A1**, scale bar: 500 μm) and the VC (**A3**, scale bar: 1000 μm; **A4**, an enlargement of the boxed area in **A3**, scale bar: 200 μm). MGV, MGD, and MGM are abbreviations for the ventral, dorsal, and medial parts of the medial geniculate nucleus, respectively. (**B**) Left: schematic drawing of the experimental setup. AAV9-syn-ChrimsonR-tdtomato was injected in the VC. The recording of field excitatory postsynaptic potential (fEPSP) was performed across layer II/III to layer V. L, laser fiber; R, recording electrode. Right: representative images of the injection site in the VC (1) and the projection terminals in the AC (2). Blue, Nissl staining. Scale bars: 1, 200 μm; 2, 20 μm. (**C**) Illustration of the fEPSP slope measurement (upper, for details refer to 'Materials and ethods') and the input/output curve of fEPSP slope evoked by laser stimulation at different intensities (bottom). Blue and yellow lines indicate 50 and 75% of fEPSP saturation. (**D**) Upper: modified high-frequency stimulation (HFS) protocol. Bottom: normalized fEPSP$_{VC-to-AC}$ slopes before and after HFS laser of VC-to-AC projection alone; inset, example traces before (dashed) and after (solid) HFS laser. Error bars represent SEM. Paired $t$-test, $t(9) = 0.878$, n.s. $p=0.403$, n = 10. (**E**) The protocol of pairing laser stimulation-evoked presynaptic input with noise-induced postsynaptic firing. (**F**) Peristimulus time histogram (PSTH)s of spike responses to noises at different sound intensities (40–90 dB, from bottom to top). (**G**) Normalized fEPSP$_{VC-to-AC}$ slopes (bottom) and example traces (upper) before and after Pre/Post Pairing with CCK-8S (red) or artificial cerebrospinal fluid (ACSF, gray) infusion in the AC. Error bars represent SEM. ****p<0.0001, n = 14 for the CCK group, n = 15 for the ACSF group, two-way repeated measures (RM) ANOVA with Bonferroni's post hoc test. See ***Supplementary file 1*** for detailed statistics.

The online version of this article includes the following source data and figure supplement(s) for figure 1:

**Source data 1.** Data for ***Figure 1*** and ***Figure 1—figure supplement 1***.

**Figure supplement 1.** VC-to-AC projection was not potentiated in the absence of any one of the following three conditions: presynaptic activation, postsynaptic firing, and CCK.

we chose the intensity that evoked reliable firing for Pre/Post Pairing. After 80 trials of only Pre/Post Pairing, the VC-to-AC inputs were not potentiated (Pre/Post Pairing, *Figure 1G*, gray, ACSF group).

In the previous studies, we have shown that CCK has an important role in neocortical plasticity (*Chen et al., 2019*; *Li et al., 2014*; *Zhang et al., 2020*). We then examined whether the VC-to-AC inputs could be potentiated after Pre/Post Pairing in the presence of CCK (CCK + Pre/Post Pairing). In line with our hypothesis, the VC-to-AC inputs were strongly potentiated after infusion of CCK-8S compared with ACSF. The averaged $fEPSP_{VC-to-AC}$ slope increased immediately after pairing and remained elevated for 1 hr in the CCK injection group (*Figure 1G*, *Figure 1—figure supplement 1E*, two-way repeated measures [RM] ANOVA, $F_{(1,27)} = 25.125$, significant interaction, p<0.0001; red, CCK before [101.3 ± 0.8%] vs. CCK after [136.5 ± 7.7%], 95% CI of increase [23.5% to 46.9%], Bonferroni's pairwise comparison, p<0.0001, n = 14; gray, ACSF before [99.9 ± 0.8%] vs. ACSF after [95.3 ± 2.9%], 95% CI of difference [–6.7% to 15.9%], Bonferroni's pairwise comparison, p=0.411, n = 15, see *Supplementary file 1* for detailed statistics). But there was no significant potentiation of the VC-to-AC inputs in the following scenarios: (i) when CCK was applied alone without pairing (CCK alone, *Figure 1—figure supplement 1F and G*, n = 8, before [104.2 ± 3.4%] vs. after [102.8 ± 9.1%], 95% CI of difference, [–25.2% to 22.5%], paired *t*-test, t(7) = 0.1357, p=0.8959, see *Supplementary file 1* for detailed statistics), (ii) when CCK was applied followed by presynaptic activation (CCK + Pre, *Figure 1—figure supplement 1H and I*, n = 10, before [99.8 ± 1.6%] vs. after [107.7 ± 5.9%], 95% CI of difference [–5.4% to 21.3%], paired *t*-test, t(9) = 1.347, p=0.2110, see *Supplementary file 1* for detailed statistics), and (iii) when CCK was applied followed by noise-induced postsynaptic firing (CCK + Post, *Figure 1—figure supplement 1J and K*, n = 10, before [103.3 ± 2.6%] vs. after [108.7 ± 8.6%], 95% CI of difference [–15.9% to 26.7%], paired *t*-test, t(9) = 0.5719, p=0.5814, see *Supplementary file 1* for detailed statistics). These findings collectively suggest that a visuo-auditory association in the AC is enabled through a direct projection from the VC-to-AC, specifically when CCK is combined with Pre/Post Pairing protocols. Conversely, the absence of any one of these three components – CCK, presynaptic, or postsynaptic activation – fails to potentiate the VC-to-AC inputs.

## HFS laser of EC-to-AC CCK⁺ terminals results in LTP of VC-to-AC inputs after pairing with postsynaptic firing in the AC evoked by noise stimulus

We have shown that cortical projection neurons in the EC mostly are CCK⁺ and glutamatergic, and that HFS induces CCK release in the AC (*Chen et al., 2019*). We then explored whether endogenous CCK could enable the potentiation of the VC-to-AC inputs. Utilizing Chronos and ChrimsonR (*Klapoetke et al., 2014*), we successfully manipulated two distinct neural pathways using dual-color activation: 473 nm wavelength light for Chronos and 635 nm for ChrimsonR (*Figure 2—figure supplement 1A and B*). We injected AAV9-Ef1α-Flex-Chronos-GFP and AAV9-hSyn-ChrimsonR-tdTomato in the EC and VC of the $Cck^{Ires-Cre}$ mouse to activate the EC-to-AC CCK⁺ projection and the VC-to-AC projection, respectively (*Figure 2A*). HFS laser (80 Hz, 5 ms/pulse, 10 pulses) of the EC-to-AC CCK⁺ projection was applied, and after a 10 ms interval, Pre/Post Pairing was followed (*Figure 2C*, HFS laser EC-to-AC + Pre/Post Pairing and *Figure 2—figure supplement 1C*). This protocol was repeated for five trials with an ITI of 10 s. Likewise, we injected AAV9-Ef1α-Flex-Chronos-GFP in the VC of CaMKIIa-Cre mice (*Figure 2B*) and applied five trials of HFS laser of the VC-to-AC projection followed by Pre/Post Pairing (*Figure 2D*, HFS laser VC-to-AC + Pre/Post Pairing) as a control.

Potentiation of the VC-to-AC inputs was observed after HFS laser EC-to-AC + Pre/Post Pairing but not after HFS laser VC-to-AC + Pre/Post Pairing (*Figure 2E*, LTP curves; *Figure 2F*, fEPSP traces; *Figure 2G*, two-way RM ANOVA, $F_{(1,21)} = 10.490$, significant interaction, p=0.004; red, HFS laser EC-to-AC + Pre/Post Pairing before [99.9 ± 1.5%] vs. after [120.7 ± 4.0%], 95% CI of increase [10.9% to 30.8%], Bonferroni's pairwise comparison, p<0.001, n = 13; green, HFS laser VC-to-AC + Pre/Post Pairing before [99.9 ± 2.0%] vs. after [97.2 ± 6.7%], 95% CI of difference [–8.6% to 14.1%], Bonferroni's pairwise comparison, p=0.623, n = 10; HFS laser EC-to-AC + Pre/Post Pairing after vs. HFS laser VC-to-AC + Pre/Post Pairing after, 95% CI of difference [8.1% to 38.9%], Bonferroni's pairwise comparison, p=0.005, see *Supplementary file 1* for detailed statistics). Additionally, this potentiation did not occur when we applied high-intensity noise stimuli (90 dB SPL) before Pre/Post Pairing (High-Intensity Noise + Pre/Post Pairing, *Figure 2—figure supplement 1D and E*, n = 10, before [102.3 ± 1.2%] vs. after [106.2 ± 5.3%], 95% CI of difference [–10.0% to 17.8%], paired *t*-test, t(9) = 0.6357,

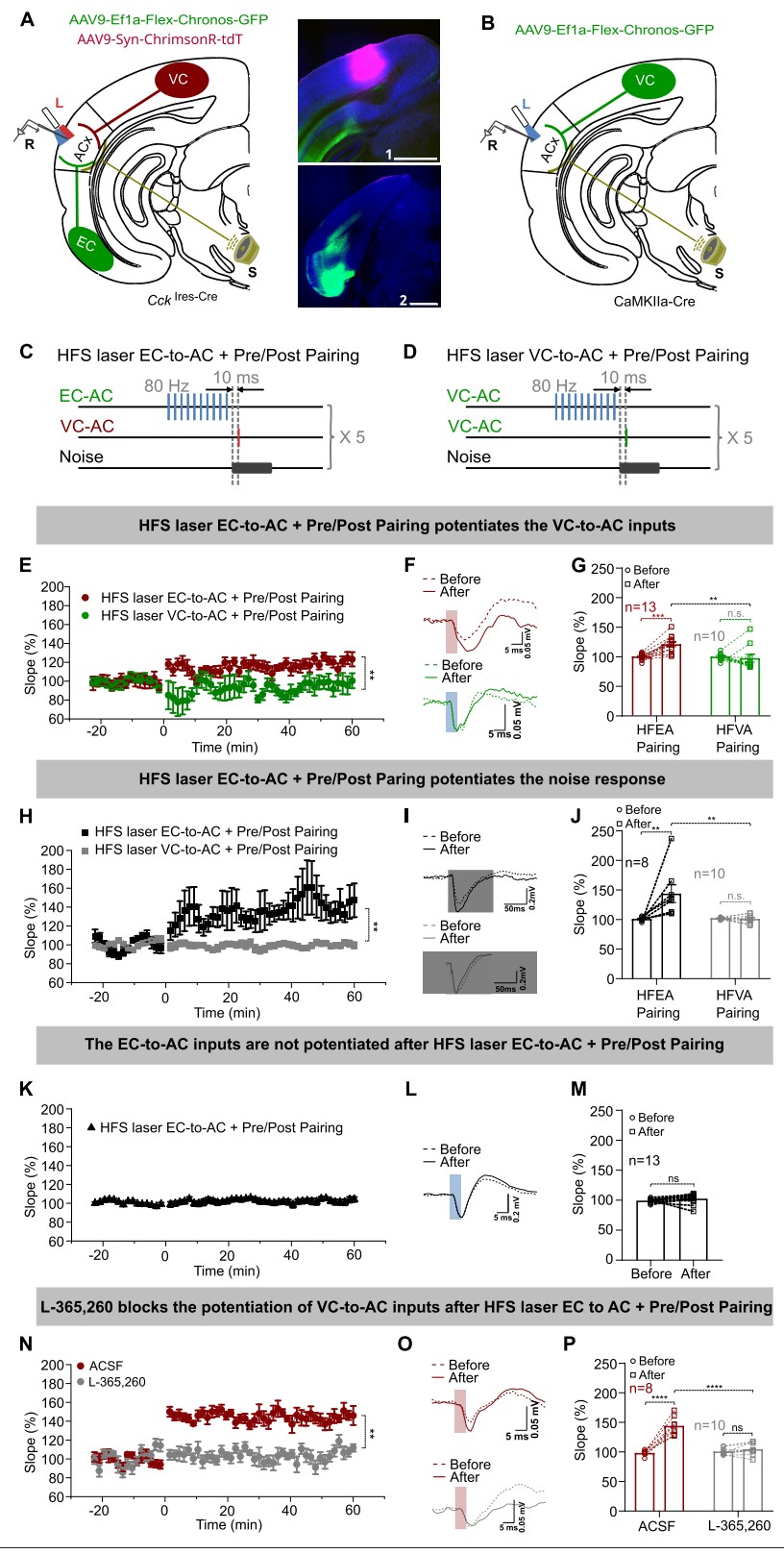

**Figure 2.** High-frequency stimulation (HFS) laser of entorhinal cortex-to-auditory cortex (EC-to-AC) CCK[+] projection but not the visual cortex-to-auditory cortex (VC-to-AC) projection induced the potentiation of VC-to-AC inputs after pairing with noise-evoked postsynaptic activation. (**A**) Left: schematic drawing of the experimental setup. AAV9-Ef1α-Flex-Chronos-GFP and AAV9-Syn-ChrimsonR-tdTomato were injected in the EC and VC of

*Figure 2 continued on next page*

*Figure 2 continued*

*Cck^Ires-Cre* mice, respectively. L, laser fiber; R, recording electrode; S, sound. Right: representative images of the injection sites in the VC (1) and the EC (2). Blue, Nissl staining. Scale bars: 1, 1000 μm; 2, 1000 μm. (**B**) Schematic drawing of the experimental setup. AAV9-Ef1α-Flex-Chronos-GFP was injected in the VC of CaMKIIa-cre mice. L, laser fiber; R, recording electrode; S, sound. (**C, D**) Protocols of HFS laser EC-to-AC + Pre/Post Pairing and HFS laser VC-to-AC + Pre/Post Pairing, respectively. (**E**) Normalized fEPSP$_{VC-to-AC}$ slopes before and after HFS laser EC-to-AC + Pre/Post Pairing (red) or HFS laser VC-to-AC + Pre/Post Pairing (green). **p<0.01, two-way repeated measures (RM) ANOVA with post hoc Bonferroni test. (**F**) Example fEPSP$_{VC-to-AC}$ traces before and after HFS laser EC-to-AC + Pre/Post Pairing (red) or HFS laser VC-to-AC + Pre/Post Pairing (green). Scale bars: upper, 5 ms and 0.05 mV; bottom, 5 ms and 0.05 mV. (**G**) Individual and average fEPSP$_{VC-to-AC}$ slope changes before and after HFS laser EC-to-AC + Pre/Post Pairing (red) or HFS laser VC-to-AC + Pre/Post Pairing (green). **p<0.01, ***p<0.001, n.s. p=0.623, n = 13 for HFS laser EC-to-AC + Pre/Post Pairing, n = 10 for HFS laser VC-to-AC + Pre/Post Pairing group, two-way RM ANOVA with post hoc Bonferroni test. (**H**) Normalized fEPSP$_{Noise}$ slopes before and after HFS laser EC-to-AC + Pre/Post Pairing (black) or HFS laser VC-to-AC + Pre/Post Pairing (gray). **p<0.01, two-way RM ANOVA with post hoc Bonferroni test. (**I**) Example fEPSP$_{Noise}$ traces before and after HFS laser EC-to-AC + Pre/Post Pairing (black) or HFS laser VC-to-AC + Pre/Post Pairing (gray). Scale bars: upper, 50 ms and 0.2 mV; bottom, 50 ms and 0.2 mV. (**J**) Individual and average fEPSP$_{Noise}$ slope changes before and after HFS laser EC-to-AC + Pre/Post Pairing (black) or HFS laser VC-to-AC + Pre/Post Pairing (gray). **p<0.01, n.s. p=0.898, n = 8 for HFS laser EC-to-AC + Pre/Post Pairing group, n = 10 for HFS laser VC-to-AC + Pre/Post Pairing group, two-way RM ANOVA with post hoc Bonferroni test (**K**) Normalized fEPSP$_{EC-to-AC}$ slopes before and after HFS laser EC-to-AC + Pre/Post Pairing. (**L**) Example fEPSP$_{EC-to-AC}$ traces before and after HFS laser EC-to-AC + Pre/Post Pairing. Scale bars: upper, 5 ms and 0.2 mV. (**M**) Individual and average fEPSP$_{EC-to-AC}$ slope changes before and after HFS laser EC-to-AC + Pre/Post Pairing. paired *t*-test, $t(12) = -1.424$, n.s. p=0.180, n = 13. (**N**) Normalized fEPSP$_{VC-to-AC}$ slopes before and after HFS laser EC-to-AC + Pre/Post Pairing in the presence of ACSF (red) or L-365,260 (gray). **p<0.01, two-way RM ANOVA with post hoc Bonferroni test. (**O**) Example fEPSP$_{VC-to-AC}$ traces before and after HFS laser EC-to-AC + Pre/Post Pairing in the presence of ACSF (red) or L-365,260 (gray). Scale bars: upper, 5 ms and 0.05 mV; bottom, 5 ms and 0.05 mV. (**P**) Individual and average fEPSP$_{VC-to-AC}$ slope changes before and after HFS laser EC-to-AC + Pre/Post Pairing in the presence of ACSF (red) or L-365,260 (gray). ****p<0.0001, n.s. p=0.6354, n = 8 for the ACSF group, n = 10 for the L-365,260 group, two-way RM ANOVA with post hoc Bonferroni test. See **Supplementary file 1** for detailed statistics.

The online version of this article includes the following source data and figure supplement(s) for figure 2:

**Source data 1.** Data for *Figure 2* and *Figure 2—figure supplement 1*.

**Figure supplement 1.** Essential trio for VC-to-AC projection potentiation: presynaptic activation, postsynaptic activation, and HFS of EC-to-AC CCK⁺ projection that induced CCK release in the AC.

---

p=0.5408, see **Supplementary file 1** for detailed statistics). fEPSPs evoked by noise stimuli were also potentiated after HFS laser EC-to-AC + Pre/Post Pairing but not HFS laser VC-to-AC + Pre/Post Pairing (*Figure 2H*, LTP curves; *Figure 2I*, fEPSP traces; *Figure 2J*, two-way RM ANOVA, $F_{(1,16)} = 9.711$, significant interaction, p=0.007; black, HFS laser EC-to-AC + Pre/Post Pairing before [100.4 ± 1.2%] vs. after [143.2 ± 14.9%], 95% CI of increase [20.5% to 65.2%], Bonferroni's pairwise comparison, p=0.001, n = 8; gray, HFS laser to VC-to-AC + Pre/Post Pairing before [101.7 ± 0.5%] vs. after [100.5 ± 2.3%], 95% CI of difference [−18.8% to 21.2%], Bonferroni's pairwise comparison, p=0.898, n = 10; HFS laser EC-to-AC + Pre/Post Pairing after vs. HFS laser VC-to-AC + Pre/Post Pairing after, 95% CI of difference [14.2% to 71.3%], Bonferroni's pairwise comparison, p=0.006, see **Supplementary file 1** for detailed statistics), and neither potentiated after High-Intensity Noise + Pre/Post Pairing (*Figure 2—figure supplement 1F and G*, n = 12, before [96.3 ± 2.1%] vs. after [98.1 ± 4.7%], 95% CI of difference [−9.0% to 12.4%], paired *t*-test, $t(11) = 0.3529$, p=0.7308, see **Supplementary file 1** for detailed statistics). However, the EC-to-AC CCK⁺ inputs were not significantly potentiated after HFS laser EC-to-AC + Pre/Post Pairing (*Figure 2K*, LTP curves; *Figure 2L*, fEPSP traces; *Figure 2M*, paired *t*-test, $t(12) = -1.424$, before [99.3 ± 0.9%] vs. after [102.7 ± 2.3%], 95% CI of difference [−8.4% to 1.8%], p=0.180, n = 13, see **Supplementary file 1** for detailed statistics). These results suggest that applying HFS laser to the EC-to-AC CCK⁺ projection, but not to the VC-to-AC CaMKII+ projection, is necessary to induce potentiation of VC-to-AC inputs. Generalized high-intensity stimuli also do not lead to the formation of potentiation. Taken together, our results demonstrate a typical form of heterosynaptic plasticity, in which the potentiation of the VC-to-AC input is not dependent on HFS activation of its own pathway but requires HFS activation of the EC-to-AC projection that presumably triggers CCK release.

To gain a deeper insight into the heterosynaptic plasticity of the VC-to-AC inputs, dependent on the EC-to-AC CCK+ pathway, we carried out various control experiments. The results indicated that the VC-to-AC inputs were not significantly potentiated in the following scenarios: (i) when HFS laser of the EC-to-AC CCK+ projection was applied alone (HFS laser EC-to-AC alone, *Figure 2—figure supplement 1H and I*, n = 12, before [99.4 ± 1.2%] vs. after [110.8 ± 6.9%], 95% CI of difference [–3.9% to 26.8%], paired *t*-test, $t(11) = 1.644$, p=0.1283, see *Supplementary file 1* for detailed statistics), (ii) when HFS laser of the EC-to-AC CCK+ projection was applied followed by presynaptic activation of the VC-to-AC inputs (HFS laser EC-to-AC + Pre, *Figure 2—figure supplement 1L and M*, n = 9, before [100.0 ± 1.3%] vs. after [103.2 ± 3.3%], 95% CI of difference [–5.0% to 11.4%], paired *t*-test, $t(8) = 0.8988$, p=0.3950, see *Supplementary file 1* for detailed statistics), and (ii) when HFS laser of the EC-to-AC CCK+ projection was applied followed by noise-induced postsynaptic firing (HFS laser EC-to-AC + Post, *Figure 2—figure supplement 1P and Q*, n = 12, before [100.6 ± 1.4%] vs. after [103.3 ± 6.9%], 95% CI of difference [–11.7% to 17.0%], paired *t*-test, $t(11) = 0.4089$, p=0.6904, see *Supplementary file 1* for detailed statistics). At the same time, fEPSPs evoked by noise stimuli were significantly potentiated after HFS laser EC-to-AC + Post (*Figure 2—figure supplement 1R and S*, n = 10, before [99.0 ± 2.3%] vs. after [109.7 ± 4.1%], 95% CI of increase [3.8% to 17.8%], paired *t*-test, $t(9) = 3.491$, p=0.0068, see *Supplementary file 1* for detailed statistics), but not after HFS laser EC-to-AC alone (*Figure 2—figure supplement 1J and K*, n = 8, before [103.1 ± 1.5%] vs. after [74.3 ± 2.7%], 95% CI of difference [–33.2% to –24.4%], paired *t*-test, $t(7) = 15.33$, p<0.0001, see *Supplementary file 1* for detailed statistics, indicating a decrease) and HFS laser EC-to-AC + Pre (*Figure 2—figure supplement 1N and O*, n = 9, before [99.7 ± 0.6%] vs. after [93.6 ± 4.0%], 95% CI of difference [–15.7% to 3.6%], paired *t*-test, $t(8) = 1.456$, p=0.1836, see *Supplementary file 1* for detailed statistics). These findings suggest that HFS laser EC-to-AC CCK+ projection, presynaptic activation, and postsynaptic firing are three prerequisites to potentiate the VC-to-AC inputs. Together with the results in *Figure 1* and *Figure 1—figure supplement 1*, we hypothesized that the HFS laser EC-to-AC CCK+ projection induced endogenous CCK release.

To validate our hypothesis, we injected AAV9-Syn-Flex-ChrimsonR-tdTomato into the EC of *Cck*Ires-Cre mice and AAV9-syn-Cck2.3 into the AC to express G protein-coupled receptor activation-based CCK sensors (*Wang et al., 2023*). This setup allowed us to use fiber photometry to effectively monitor CCK dynamics in the AC when HFS laser was applied to the CCK+ neurons in the EC (*Figure 2—figure supplement 1T and U*). The results demonstrated that HFS laser of the EC CCK+ neurons evoked a significant rising of CCK signal in the AC (*Figure 2—figure supplement 1V and W*, two-way RM ANOVA, $F_{(1, 16)} = 4.876$, significant interaction, p=0.0422; CCK sensor before [0.0 ± 0.0] vs. after [2.6 ± 1.1], 95% CI of increase [0.5 to 4.6], Bonferroni's pairwise comparison, p=0.0125, n = 9; isosbestic before [0.0 ± 0.0] vs. after [0.0 ± 0.1], 95% CI of difference [–2.0 to 2.0], Bonferroni's pairwise comparison, p>0.9999, n = 9; see *Supplementary file 1* for detailed statistics), indicating that endogenous CCK is released in the AC in response to HFS laser stimulation of the CCK+ neurons in the EC. Furthermore, to assess the role of endogenously released CCK in the potentiation of the VC-to-AC inputs, we infused L-365,260, a CCK B receptor antagonist, into the AC prior to the HFS laser EC-to-AC + Pre/Post Pairing protocol. We observed that L-365,260 blocked the potentiation of the VC-to-AC inputs following this protocol, whereas ACSF did not show this effect (*Figure 2N*, LTP curves; *Figure 2O*, fEPSP traces; *Figure 2P*, two-way RM ANOVA, $F_{(1,16)} = 53.29$, significant interaction, p<0.0001; red, ACSF before [98.3 ± 4.3%] vs. after [144.1 ± 4.3%], 95% CI of increase [35.2% to 56.4%], Bonferroni's pairwise comparison, p<0.0001, n = 8; gray, L-365,260 before [100.6 ± 3.8%] vs. after [104.6 ± 3.8%], 95% CI of difference [–13.4% to 5.5%], Bonferroni's pairwise comparison, p=0.6354, n = 10; ACSF after vs. L-365,260 after, 95% CI of difference [28.8% to 50.2%], Bonferroni's pairwise comparison, p<0.0001, see *Supplementary file 1* for detailed statistics).

## HFS laser of the EC-to-AC CCK+ terminals ex vivo leads to LTP of the VC-to-AC inputs after pairing with electrical stimulation

We also performed similar experiments at the single-cell level ex vivo. Slices were prepared from *Cck*Ires-Cre mice after injection of AAV9-Ef1α-Flex-Chronos-GFP in the EC and of AAV9-Syn-ChrimsonR-tdTomato in the VC (*Figure 3A*). Pyramidal neurons in the AC were patched (*Figure 3B*), and excitatory postsynaptic currents (EPSCs) evoked by laser stimulation of the VC-to-AC projection (EPSCVC-to-AC, *Figure 3C*) and ES of the AC (EPSCES, *Figure 3D*) were recorded. HFS laser of the EC-to-AC CCK+

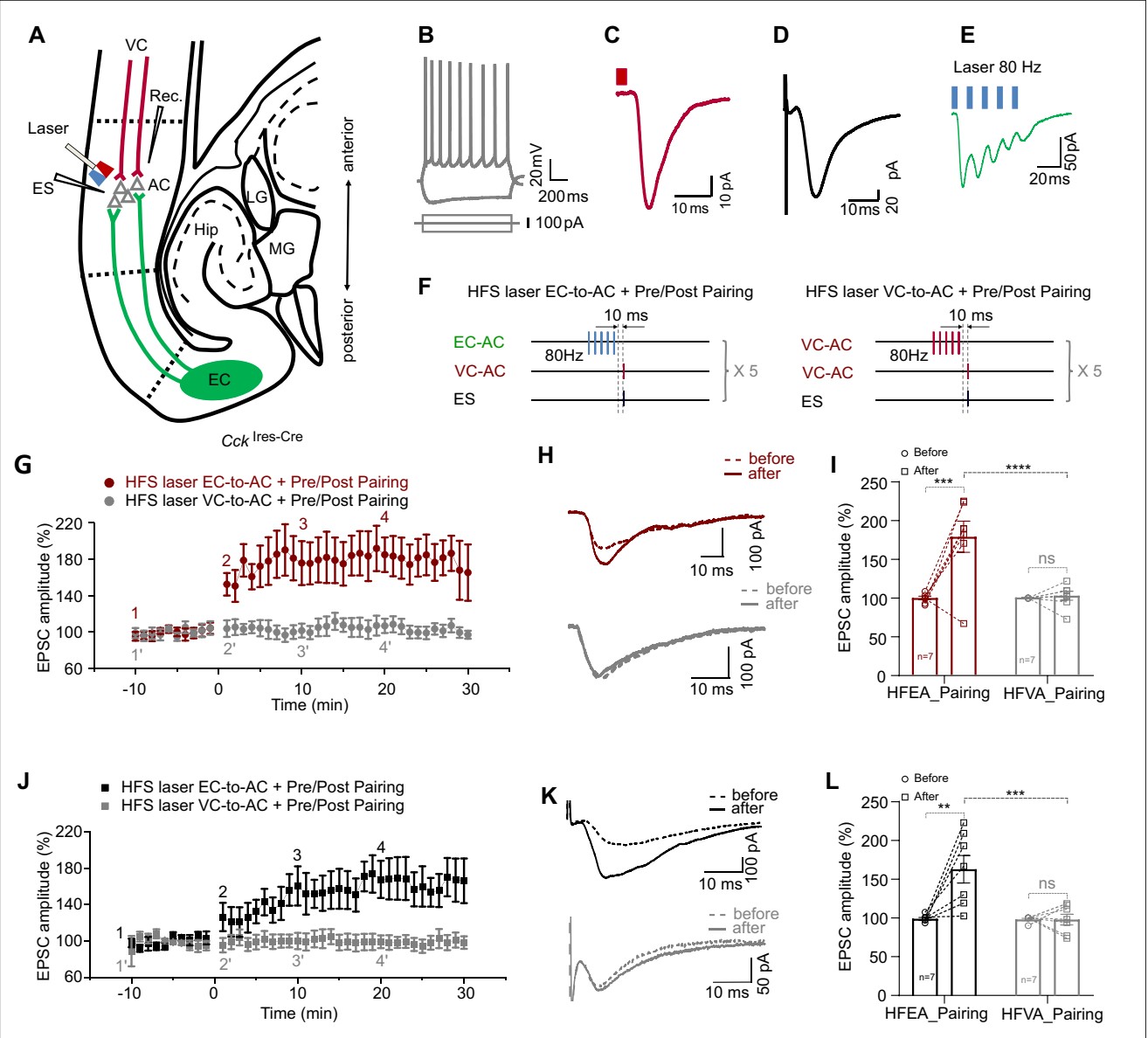

**Figure 3.** High-frequency stimulation (HFS) laser of entorhinal cortex-to-auditory cortex (EC-to-AC) CCK+ projection ex vivo leads to long-term potentiation (LTP) of visual cortex-to-auditory cortex (VC-to-AC) inputs after pairing with electrical stimulation (ES). (**A**) Positions of the whole-cell recording pipette, electrical stimulation electrode, and the optical fiber in a slice of *Cck*Ires-Cre mice with AAV-Ef1α-Flex-Chronos-GFP injected in the EC and AAV-Syn-ChrimsonR-tdTomato injected in the VC. (**B**) Representative pyramidal neuron firing in response to current injection. (**C, D**) Representative traces of EPSC$_{VC-to-AC}$ (**C**) and EPSC$_{ES}$ (**D**) of pyramidal neuron. EPSC, excitatory postsynaptic current. Scale bars: (**C**), 10 ms and 10 pA; (**D**), 10 ms and 20 pA. (**E**) Representative EPSC trace in response to HFS laser of the EC-to-AC CCK+ terminals (blue rectangles, 80 Hz, 5 ms/pulse). (**F**) Protocols of HFS laser EC-to-AC + Pre/Post Pairing (left) and HFS laser VC-to-AC + Pre/Post Pairing (right). (**G**) Normalized EPSC$_{VC-to-AC}$ amplitudes before and after HFS laser EC-to-AC + Pre/Post Pairing (red) or HFS laser VC-to-AC + Pre/Post Pairing (gray). (**H**) Example EPSC$_{VC-to-AC}$ traces before (dashed, at timepoint 1 or 1' in **G**) and after (solid, at timepoint 4 or 4' in **G**) HFS laser EC-to-AC + Pre/Post Pairing (red, upper) or HFS laser VC-to-AC + Pre/Post Pairing (gray, bottom). (**I**) Individual and average EPSC$_{VC-to-AC}$ amplitude changes before and after HFS laser EC-to-AC + Pre/Post Pairing (red) or HFS laser VC-to-AC + Pre/Post Pairing (gray). ***p<0.001, ****p<0.0001, n.s. p>0.9999, n = 7 for both HFS laser EC-to-AC + Pre/Post Pairing group and HFS laser VC-to-AC + Pre/Post Pairing group, two-way repeated measures (RM) ANOVA with post hoc Bonferroni test. (**J**) Normalized EPSC$_{ES}$ amplitudes before and after HFS laser EC-to-AC + Pre/Post Pairing (black) or HFS laser VC-to-AC + Pre/Post Pairing (gray). (**K**) Example EPSC$_{ES}$ traces before (dashed, at timepoint 1 or 1' in **J**) and after (solid, at timepoint 4 or 4' in **J**) HFS laser EC-to-AC + Pre/Post Pairing (black, upper) or HFS laser VC-to-AC + Pre/Post Pairing (gray, bottom). (**L**) Individual and average EPSC$_{ES}$ amplitude changes before and after HFS laser EC-to-AC + Pre/Post Pairing (black) or HFS laser VC-to-AC + Pre/Post Pairing (gray). **p<0.01, ***p<0.001, n.s. p>0.9999, n = 7 for both HFS laser EC-to-AC + Pre/Post Pairing group and HFS laser VC-to-AC + Pre/Post Pairing group, two-way RM ANOVA with post hoc Bonferroni test. See **Supplementary file 1** for detailed statistics.

The online version of this article includes the following source data and figure supplement(s) for figure 3:

*Figure 3 continued on next page*

*Figure 3 continued*

**Source data 1.** Data for *Figure 3*.

**Figure supplement 1.** Examples of EPSC traces evoked by different stimuli at specified time points in *Figure 3* recorded under various conditions.

projection (*Figure 3E*) was followed by the pairing of presynaptic activation evoked by laser stimulation of the VC-to-AC projection and postsynaptic activation evoked by ES, which was repeated five times with an ITI of 10 s (HFS laser EC-to-AC + Pre/Post Pairing, *Figure 3F*, left). As a control, we replaced the HFS laser of EC-to-AC CCK$^+$ projections with HFS laser of VC-to-AC projection (HFS laser VC-to-AC + Pre/Post Pairing, *Figure 3F*, right).

Similar to the in vivo results, the amplitude of EPSC$_{VC-to-AC}$ significantly increased after HFS laser EC-to-AC + Pre/Post Pairing but not after HFS laser VC-to-AC + Pre/Post Pairing (*Figure 3G*, LTP curves; *Figure 3H*, EPSC$_{VC-to-AC}$ traces; *Figure 3I*, two-way RM ANOVA, F$_{(1,12)}$ = 13.16, significant interaction, p=0.0035; red, increased by 78.9 ± 15.0% after HFS laser EC-to-AC + Pre/Post Pairing, 95% CI of increase [40.6% to 117.2%], Bonferroni's pairwise comparison, p=0.004, n = 7; gray, changed by 2.2 ± 15.0% after HFS laser VC-to-AC + Pre/Post Pairing, 95% CI of difference [-36.1% to 40.5%], Bonferroni's pairwise comparison, p>0.9999, n = 7; HFS laser EC-to-AC + Pre/Post Pairing after vs. HFS laser VC-to-AC + Pre/Post Pairing after, 76.1 ± 14.9% of difference, 95% CI of difference [40.6% to 111.7%], Bonferroni's pairwise comparison, p<0.0001, see *Supplementary file 1* for detailed statistics; *Figure 3—figure supplement 1A and B*, 10 successive example traces and their averaged trace of the EPSCs at different timepoints as shown in *Figure 3G*). Likewise, the EPSC$_{ES}$ amplitude significantly increased after HFS laser EC-to-AC + Pre/Post Pairing but not after HFS laser VC-to-AC + Pre/Post Pairing (*Figure 3J*, LTP curves; *Figure 3K*, EPSC$_{ES}$ traces; *Figure 3L*, two-way RM ANOVA, F$_{(1,12)}$ = 10.54, significant interaction, p=0.0070; black, HFS laser EC-to-AC + Pre/Post Pairing before vs. after, increased by 64.0 ± 14.0%, 95% CI of increase [28.3% to 99.8%], Bonferroni's pairwise comparison, p=0.0013, n = 7; gray, changed by 0 ± 14.0% after HFS laser VC-to-AC + Pre/Post Pairing, 95% CI of difference [−35.65% to 35.86%], Bonferroni's pairwise comparison, p>0.9999, n = 7; HFS laser EC-to-AC + Pre/Post Pairing after vs. HFS laser VC-to-AC + Pre/Post Pairing after, 65.0 ± 13.6% of difference, 95% CI of difference [32.5% to 97.4%], Bonferroni's pairwise comparison, p=0.0001, see *Supplementary file 1* for detailed statistics; *Figure 3—figure supplement 1C and D*, 10 successive traces and their averaged trace at different timepoints as shown in *Figure 3J*).

## EC-to-AC-projecting neurons have higher CCK expression levels than VC-to-AC-projecting neurons

The results from both in vivo and ex vivo experiments showed that HFS laser EC-to-AC + Pre/Post Pairing led to potentiation of the input from the VC-to-AC. In contrast, a similar protocol applied from the VC to the AC (HFS laser VC-to-AC + Pre/Post Pairing) did not induce such potentiation. We established that HFS laser stimulation of CCK$^+$ neurons in the EC resulted in CCK release in the AC, as depicted in *Figure 2—figure supplement 1T–W*. Moreover, infusing a CCKB receptor antagonist into the AC inhibited the potentiation of the VC-to-AC inputs following the HFS laser EC-to-AC + Pre/Post Pairing, as shown in *Figure 2N–P*. This leads to a consideration of the underlying differences between the EC-to-AC projection and the VC-to-AC projection. A plausible explanation could be the variation in CCK expression levels between the AC-projecting neurons in the EC and those in the VC.

We next explored whether the levels of *Cck* transcript, and thus possibly CCK peptide, could be underlying this difference by using RNAscope combined with retrograde tracing with AAV virus. We injected AAVretro-hSyn-Cre-WPRE-hGH in the AC of Ai14 mice, a Cre reporter line, retrogradely labeling the EC and VC neurons projecting to AC with Cre-dependent tdTomato. The expression level of *Cck* was then assessed by RNAscope, a semi-quantitative in situ hybridization method (*Figure 4A–D*). The RNAscope assay allows quantitative detection of RNA species and has been widely used to determine gene expression levels (*Caldwell et al., 2021*; *Jolly et al., 2019*; *Chan et al., 2018*). We found that the *expression level* of *Cck* was significantly higher among projecting neurons in the EC than in the VC across three animals analyzed (*Figure 4E*, Welch's *t*-test, *t*(539) = 7.615, EC [1.48 ± 0.05] vs. VC [1.00 ± 0.04], 95% CI of difference [0.36 to 0.60], p<0.0001, n = 345 for EC, n = 199 for VC; *Figure 4—figure supplement 1*, animal 1, Welch's *t*-test, *t*(78) = 5.315, p<0.0001; animal 2, Welch's *t*-test, *t*(51) = 2.521, p=0.015; animal 3, Welch's *t*-test, *t*(91) = 3.0122, p=0.003,

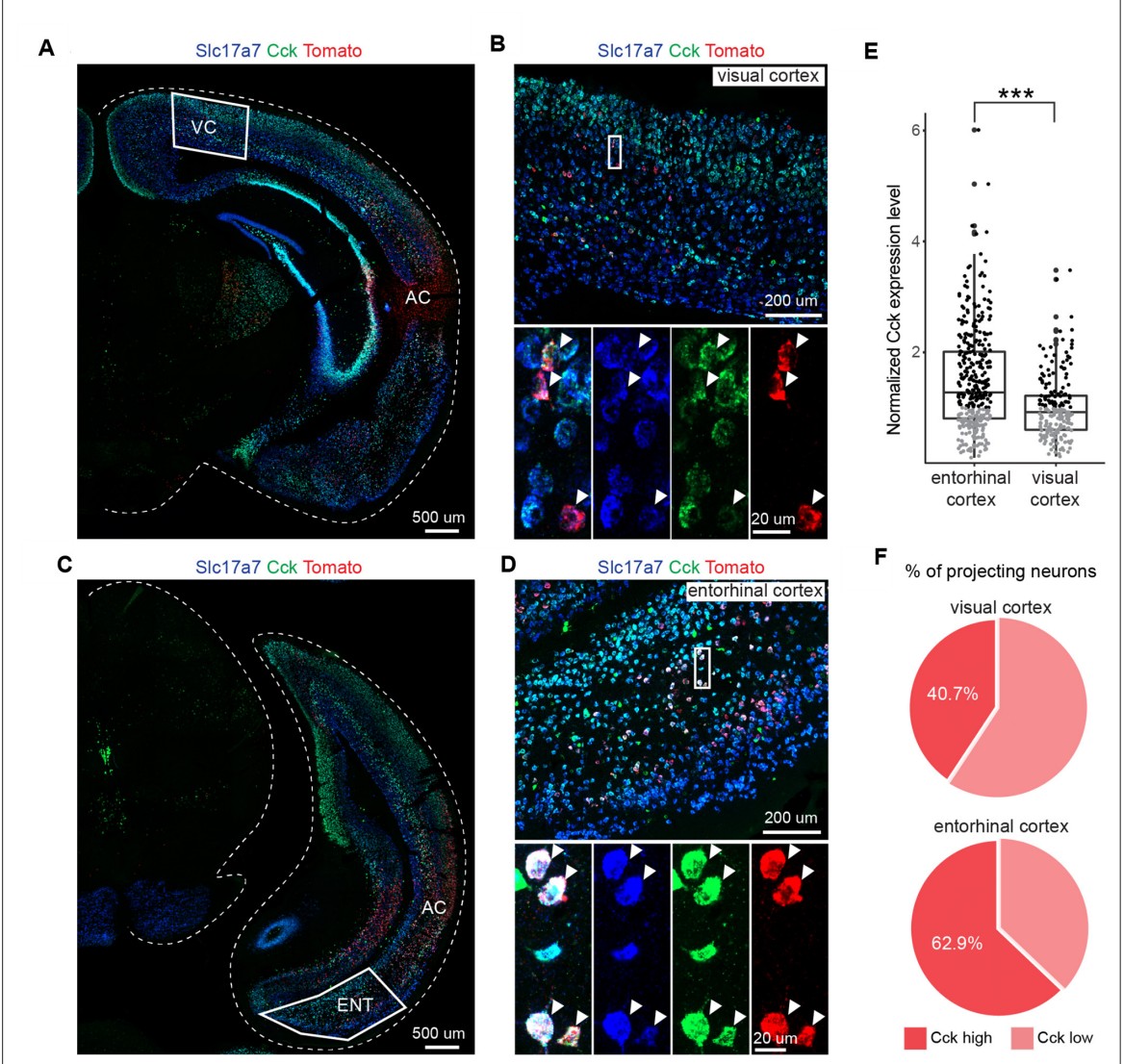

**Figure 4.** Auditory cortex (AC)-projecting neurons in the entorhinal cortex (EC) express higher level of *Cck* than those in the visual cortex (VC). (**A**) Overview of injection site at the AC and projecting neurons in the VC. Scale bar: 500 um. (**B**) Expression of *Slc17a7* (*vGlut1*) and *Cck* in retrogradely labeled neurons (tdTomato+) in the VC. Scale bars: upper, 200 μm; bottom, 20 μm. (**C**) Overview of injection site at the AC and projecting neurons in the EC. Scale bar: 500 μm. (**D**) Expression of *Slc17a7* and *Cck* in retrogradely labeled neurons (tdTomato+) in the EC. Scale bars: upper, 200 μm; bottom, 20 μm. (**E**) Comparison of *Cck* expression level in AC-projecting neurons of EC and VC (data points are from three animals). Unpaired *t*-test, ***p<0.001. Black, high level; gray, low level. (**F**) Pie chart shows percentage of projecting neurons expressing low and high *Cck* level in the VC (upper) and the EC (bottom), respectively. See *Supplementary file 1* for detailed statistics.

The online version of this article includes the following source data and figure supplement(s) for figure 4:

**Source data 1.** Data for *Figure 4* and *Figure 4—figure supplement 1*.

**Figure supplement 1.** Comparison of *Cck* expression level in neurons in the entorhinal cortex (EC) and visual cortex (VC), which project to the auditory cortex (AC).

see *Supplementary file 1* for detailed statistics). The *proportion* of projecting neurons expressing elevated *Cck* levels was also higher in the EC compared with the VC (*Figure 4F*). In terms of cell count, we identified 345 CCK⁺ AC-projecting neurons in the EC and 199 in the VC across three animals. Out of these, 217 out of 345 cells in the EC and 81 out of 199 cells in the VC showed elevated *Cck* transcripts. These results suggest that after HFS laser stimulation more CCK is released from the AC-projecting neurons in the EC than from those in the VC, which may, at least, be one explanation why the former but not the latter can produce LTP.

## Effect of different parameters of the pairing protocol on the potentiation level of VC-to-AC inputs

Neuropeptide release likely is frequency-dependent (*Bean and Roth, 1991*; *Hökfelt, 1991*; *Iverfeldt et al., 1989*; *Lundberg and Hökfelt, 1983*; *Shakiryanova et al., 2005*; *Whim and Lloyd, 1989*), and our results suggest that CCK released from the EC-to-AC CCK+ projection was critical for generating visuo-auditory cortical LTP. We hypothesized that the frequency of the laser used to stimulate the EC-to-AC CCK+ projection was critical for the level of potentiation of the VC-to-AC input. We therefore varied the *frequency of the laser stimulation* (80, 40, 10, or 1 Hz). As shown in *Figure 5A*, left, the delay between the termination of repetitive laser stimulation of the CCK+ EC-to-AC projection and presynaptic activation (Delay 1) was set at 10 ms, and the delay between pre- and postsynaptic activation (Delay 2) was set at 0 ms. The potentiation level of the VC-to-AC inputs showed a tendency to increase as the frequency of laser stimulation of the CCK+ EC-to-AC projection increased (*Figure 5A*, right, two-way RM ANOVA, $F_{(3,34)} = 10.666$, significant interaction, $p<0.001$; 1 Hz before [99.7 ± 1.3%] vs. after [96.6 ± 2.9%], 95% CI of difference [−3.6% to 9.8%], Bonferroni's pairwise comparison, p=0.352, n = 9; 10 Hz before [98.5 ± 1.2%] vs. after [105.8 ± 2.5%], 95% CI of increase [0.2% to 14.4%], Bonferroni's pairwise comparison, p=0.044, n = 8; 40 Hz before [99.4 ± 0.8%] vs. after [110.5 ± 1.8%], 95% CI of increase [4.0% to 18.3%], Bonferroni's pairwise comparison, p=0.003, n = 8; 80 Hz before [99.9 ± 1.5%] vs. after [120.7 ± 4.0%], 95% CI of increase [15.2% to 26.4%], Bonferroni's pairwise comparison, p<0.001, n = 13, see *Supplementary file 1* for detailed statistics). If higher than 10 Hz, the VC-to-AC inputs were significantly potentiated. However, at 1 Hz no significant potentiation was observed.

In contrast to small-molecule neurotransmitters that are rapidly cleared by reuptake pumps, neuropeptides are mostly released extrasynaptically, are removed/inactivated more slowly, and may have longer-lasting effects. Thus, we explored the role of Delay 1, that is, if the *time* interval between the termination of HFS laser and the Pre/Post Pairing influenced the degree of potentiation of the VC-to-AC inputs (Delay 2 = 0 ms, HFS laser frequency = 80 Hz, *Figure 5B*, left). The VC-to-AC inputs were significantly potentiated, when Delay 1 was 10, 85, 235, or 535 ms rather than 885 or −65 ms (*Figure 5B*, right, two-way RM ANOVA, $F_{(5,59)} = 7.115$, significant interaction, p<0.001; 10 ms before [99.9 ± 1.5%] vs. after [120.7 ± 4.0%], 95% CI of increase [14.6% to 27.1%], Bonferroni's pairwise comparison, p<0.001, n = 13; 85 ms before [99.8 ± 0.8%] vs. after [119.1 ± 3.5%], 95% CI of increase [13.0% to 25.5%], Bonferroni's pairwise comparison, p<0.001, n = 13; 235 ms before [99.5 ± 3.7%] vs. after [117.0 ± 5.2%], 95% CI of increase [8.3 to 26.7%], Bonferroni's pairwise comparison, p<0.001, n = 6; 535 ms before [99.4 ± 0.6%] vs. after [110.4 ± 1.9%], 95% CI of increase [3.8% to 18.0%], Bonferroni's pairwise comparison, p=0.003, n = 10; 885 ms before [99.6 ± 0.8%] vs. after [102.7 ± 1.3%], 95% CI of difference [-10.2% to 4.0%], Bonferroni's pairwise comparison, p=0.385, n = 10; −65 ms before [99.0 ± 0.8%] vs. after [100.1 ± 3.0%], 95% CI of difference [− 6.4% to 6.0%], Bonferroni's pairwise comparison, p=0.945, n = 13, see *Supplementary file 1* for detailed statistics).

The Hebbian theory states, popularly, that "cells that fire together wire together" (*Löwel and Singer, 1992*), a more accurate interpretation being 'synaptic strength increases when the presynaptic neuron always fires immediately before the postsynaptic neuron' (*Caporale and Dan, 2008*). Based on this, the interval between pre- and postsynaptic activation should be critical for potentiation. In the next experiment, the interval (Delay 2) between the VC-to-AC projection activation (i.e., presynaptic activation) and natural AC activation (i.e., postsynaptic activation) was set as the only variable (Delay 1 = 10 ms, HFS laser frequency = 80 Hz, *Figure 5C*, left). The potentiation of the VC-to-AC inputs showed a decreasing trend as Delay 2 increased. Significant potentiation was observed when Delay 2 was 0, 50, 200, rather than 400, 800 ms, and ∞ (without noise) (*Figure 5C*, right, two-way RM ANOVA, $F_{(5,51)} = 4.133$, significant interaction, p=0.003; 0 ms before [99.9 ± 1.5%] vs. after [120.7 ± 4.0%], 95% CI of increase [15.0% to 26.7%], Bonferroni's pairwise comparison, p<0.001, n = 13; 50 ms before [99.7 ± 0.6%] vs. after [110.7 ± 2.7%], 95% CI of increase [4.6% to 17.2%], Bonferroni's pairwise comparison, p=0.001, n = 11; 200 ms before [102.0 ± 1.2%] vs. after [110.6 ± 3.3%], 95% CI of increase [2.6% to 14.7%], Bonferroni's pairwise comparison, p=0.006, n = 12; 400 ms before [97.5 ± 1.5%] vs. after [105.3 ± 2.1%], 95% CI of difference [−16.4% to 0.8%], Bonferroni's pairwise comparison, p=0.073, n = 6; 800 ms before [98.6 ± 1.8%] vs. after [101.8 ± 2.1%], 95% CI of difference [− 11.8%% to 5.4%], Bonferroni's pairwise comparison, p=0.454, n = 6; ∞ before [100.0 ± 1.3%] vs. after [103.2 ± 3.3%], 95% CI of difference [− 10.2% to

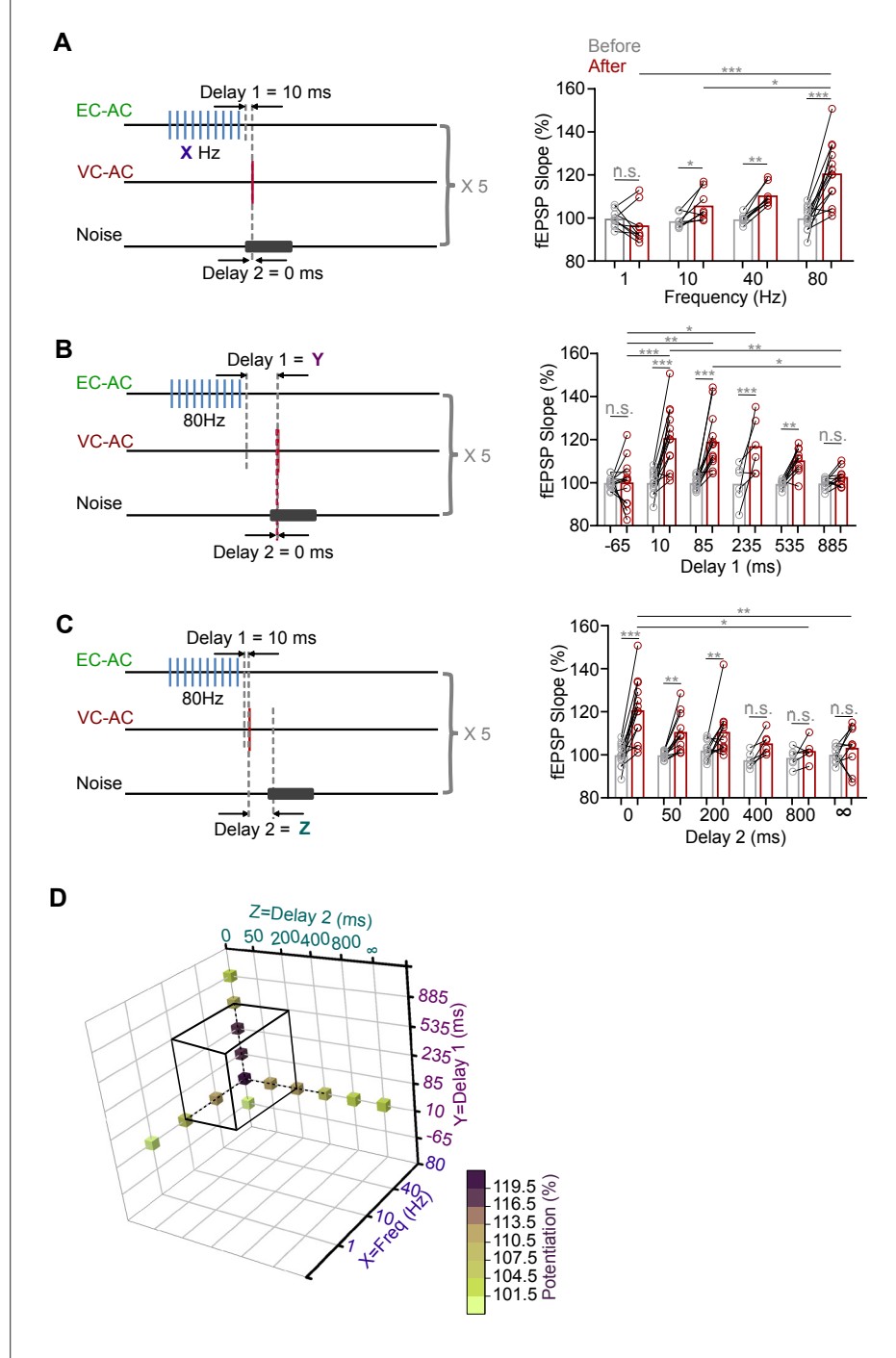

**Figure 5.** Effect of different parameters of the pairing protocol on the potentiation level of the visual cortex-to-auditory cortex (VC-to-AC) inputs. (**A**) Left: schematic drawing of experiment design. Delay 1 = 10 ms, Delay 2 = 0 ms, and the frequency was varied (80, 40, 10, or 1 Hz). Right: individual and average fEPSP$_{VC-to-AC}$ slopes (normalized to the baseline) after pairing at different frequencies. Two-way repeated measures (RM) ANOVA with post hoc Bonferroni test, n.s., no significant, *p<0.05, **p<0.01, ***p<0.001, n = 9 for 1 Hz, n = 8 for 10 Hz, n = 8 for 40 Hz, n = 13 for 80 Hz. Data points in groups 1 Hz and 40 Hz refer to our previous study (***Zhang et al., 2020***). (**B**) Left: schematic drawing of experiment design. HFS laser frequency = 80 Hz, Delay 2 = 0 ms, and Delay 1 was varied (10, 85 235, 535, 885, and –65 ms). Right: individual and average fEPSP$_{VC-to-AC}$ slopes (normalized to the baseline) after pairing at different Delay 1s. Two-way RM ANOVA with post hoc Bonferroni test, n.s., no significant, *p<0.05, **p<0.01, ***p<0.001, n = 13, 13, 13, 6, 10, 10 for Delay 1 = –65, 10, 85, 235, 535, and 885 ms, respectively. (**C**) Left:

*Figure 5 continued on next page*

*Figure 5 continued*

schematic drawing of experiment design. HFS laser frequency = 80 Hz, Delay 1 = 10 ms, and Delay 2 was varied (0, 50, 200, 400, 800 ms, and ∞). Right: individual and average fEPSP$_{VC-to-AC}$ slopes (normalized to the baseline) after pairing at different Delay 2s. Two-way RM ANOVA with post hoc Bonferroni test, n.s., not significant, *p<0.05, **p<0.01, ***p<0.001, n = 13, 11, 12, 6, 6, 9 for Delay 2 = 0, 50, 200, 400, 800 ms, and ∞, respectively. (**D**) Three-dimensional summary of the effect of different parameters (Frequency, Delay 1 and Delay 2) on the potentiation level of the VC-to-AC inputs. Parameters locate inside black cubes can induce significant potentiation. See for *Supplementary file 1* detailed statistics.fEPSP, field excitatory postsynaptic potential; HFS, high-frequency stimulation.

The online version of this article includes the following source data for figure 5:

**Source data 1.** Data for *Figure 5*.

3.8%], Bonferroni's pairwise comparison, p=0.363, n = 9, see *Supplementary file 1* for detailed statistics).

Taken together, our study indicates that significant potentiation of the VC-to-AC inputs can be observed (*Figure 5D*, black cube) across five pairing trials with a 10 s ITI, under certain tested conditions: (i) the frequency of repetitive laser stimulation of the CCK$^+$ EC-to-AC projection was maintained at 10 Hz or higher (as we did not test frequencies between 1 and 10 Hz), (ii) Delay 1 was set within the tested range of 10–535 ms (noting the absence of data between –65 and 10 ms), and (iii) Delay 2 was within the range of 0–200 ms (acknowledging that negative values for Delay 2 were not explored).

## Inactivation of the EC-to-AC CCK$^+$ projection prevents the establishment of the visuo-auditory association behaviorally

Next, we ask whether the endogenous CCK released from the EC-to-AC pathway is essential for the generation of visuo-auditory associations, which can be reflected in a behavioral context. To that end, we adopted the Designer Receptors Exclusively Activated by Designer Drugs-based chemogenetic silencing tools. We bilaterally injected AAV9-Syn-DIO-hM4Di-EYFP or AAV9-Syn-DIO-EYFP in the EC of *Cck*$^{Ires-Cre}$ mice to express the hM4Di in the CCK$^+$ neurons in the EC. Neurons expressing hM4Di can be suppressed in the presence of clozapine-N-oxide (CNO). First, CNO was injected bilaterally into the AC to inactivate the EC-to-AC CCK$^+$ pathway, followed by a 25-trial pairing session of the visual stimulus (VS) with the auditory stimulus (AS). We repeated the above drug and pairing session four times per day and on three consecutive days. On day 4, baseline tests for the freezing response to the AS and VS were performed (three trials) before the mouse was fear-conditioned to the AS. After fear conditioning, freezing responses to the AS and VS were further examined on day 5 (*Figure 6A*). As expected, mice showed no freezing response to the AS before conditioning, but a high freezing rate to the AS after conditioning (*Figure 6B*, GFP-CNO-AS-Baseline [8.8 ± 2.9%] vs. GFP-CNO-AS-Post intervention [69.0 ± 4.4%], 95% CI of difference [47.3%–73.0%], p<0.0001, n = 10; EC-AC-hM4Di-CNO-AS-Baseline [11.2 ± 3.7%] vs. EC-AC-hM4Di-CNO-AS-Post intervention [65.9 ± 4.0%], 95% CI of difference [39.3% to 70.2%], p<0.0001, n = 7; two-way RM ANOVA with post hoc Bonferroni test, see *Supplementary file 1* for detailed statistics). The GFP group showed a significantly increased freezing response to the VS, indicating that an association between the AS and VS had been established by the pairings (*Figure 6B*, blue square, GFP-CNO-VS-Baseline [7.0 ± 2.1%] vs. GFP-CNO-VS-Post intervention [44.9 ± 4.8%], 95% CI of difference [25.0% to 50.8%], p<0.0001, n = 10, two-way RM ANOVA with post hoc Bonferroni test, see *Supplementary file 1* for detailed statistics). However, inactivation of the EC-to-AC CCK$^+$ projection (EC-AC-hM4Di-CNO) blocked this association, resulting in a nil response to the VS (*Figure 6B*, beige square, EC-AC-hM4Di-CNO-VS-Baseline [6.7 ± 1.3%] vs. EC-AC-hM4Di-CNO-VS-Post intervention [8.4 ± 1.7%], 95% CI of difference [–17.2% to 13.7%], p>0.9999, n = 7, two-way RM ANOVA with post hoc Bonferroni test, see *Supplementary file 1* for detailed statistics). There was also a significant difference between the freezing rates to the VS of experimental and control groups (*Figure 6B*, EC-AC-hM4Di-CNO-VS-Post intervention [8.4 ± 1.7%, n = 7] vs. GFP-CNO-VS-Post intervention [44.9 ± 4.8%, n = 10], 95% CI of difference [20.8% to 52.1%], p<0.0001, two-way RM ANOVA with post hoc Bonferroni test, see *Supplementary file 1* for detailed statistics). These results demonstrate that inactivation of the EC-to-AC CCK$^+$ projection prevented the generation of the association between the VS and AS and suggest an essential role of endogenous CCK in the generation of the visuo-auditory association.

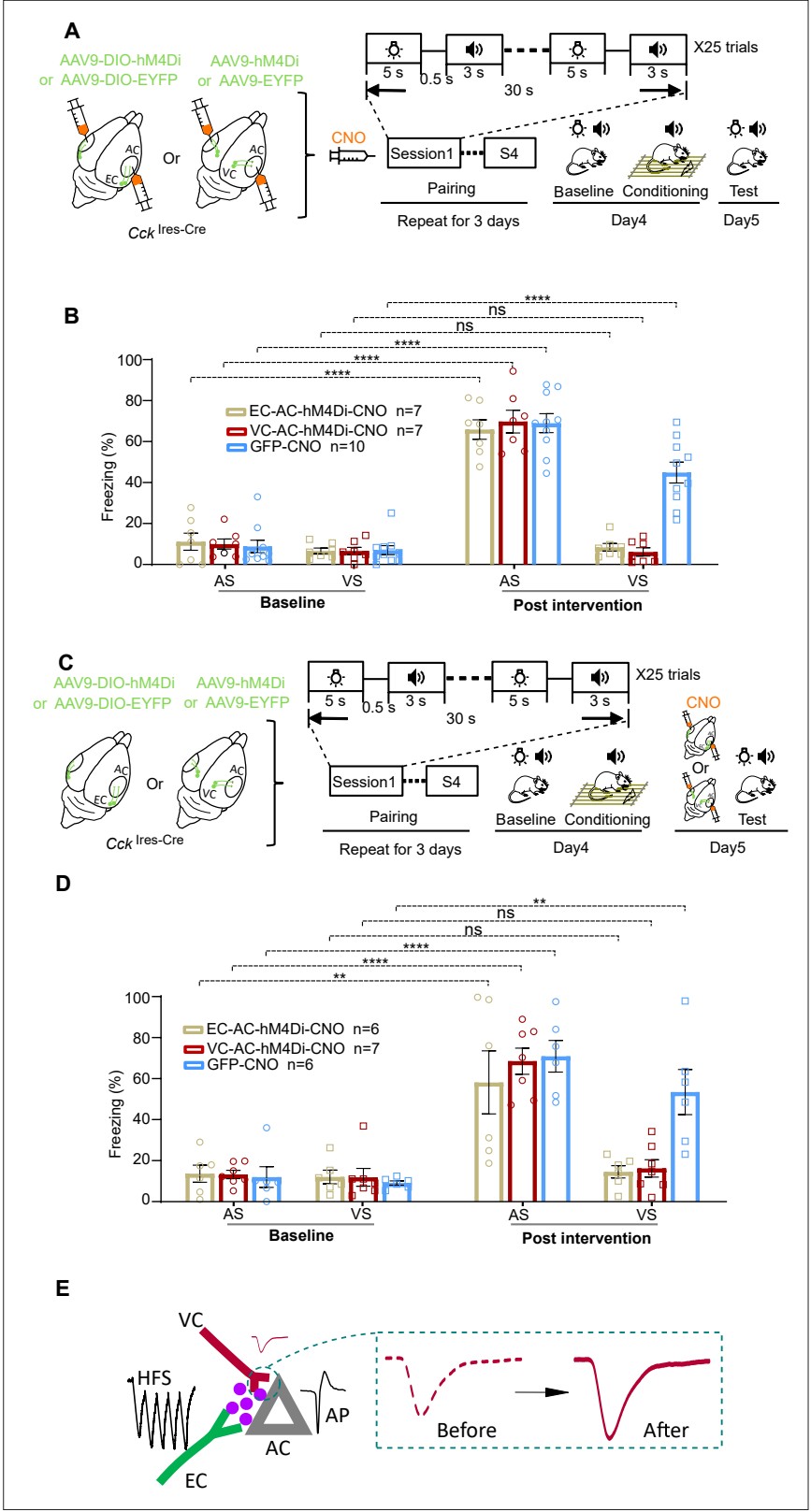

**Figure 6.** Roles of the entorhinal cortex-to-auditory cortex (EC-to-AC) CCK+ projection and visual cortex-to-auditory cortex (VC-to-AC) projection in establishing and retrieving of the visuo-auditory associative memories. (**A**) Schematic drawing of the experimental design that the chemogenetic manipulation was applied in the encoding phase. (**B**) Bar chart showing freezing percentages to the auditory stimulus (AS) and visual stimulus

*Figure 6 continued*

(VS) before and after the conditioning in different conditions. ****p<0.01, n.s., not significant, n = 7 for the EC-AC-hM4Di-CNO group, n = 7 for the VC-AC-hM4Di-CNO group, and n = 10 for the GFP-CNO group, two-way repeated measures (RM) ANOVA with post hoc Bonferroni test. (**C**) Schematic drawing of the experimental design that the chemogenetic manipulation was applied in the retrieval phase. (**D**) Bar chart showing freezing percentages to the AS and VS before and after the conditioning in different conditions. ****p<0.01, **p<0.01, n.s., not significant, n = 6 for the EC-AC-hM4Di-CNO group, n = 7 for the VC-AC-hM4Di-CNO group, and n = 6 for the GFP-CNO group, two-way RM ANOVA with post hoc Bonferroni test. (**E**) Schematic drawing of our theory that endogenous CCK, presynaptic activation, and postsynaptic firing enables the plasticity of the VC-to-AC inputs. See *Supplementary file 1* for detailed statistics.

The online version of this article includes the following source data and figure supplement(s) for figure 6:

**Source data 1.** Data for *Figure 6* and *Figure 6—figure supplement 1*.

**Figure supplement 1.** Visuo-auditory associative memory could not be formed without CCK.

In a similar strategy, we bilaterally expressed hM4Di in the VC and infused CNO in the AC to silence the VC-to-AC projection to explore its role in establishing the visuo-auditory association behaviorally (*Figure 6A*). The results indicated that inactivation of the VC-to-AC projection (VC-AC-hM4Di-CNO) also blocked the association between VS and AS (*Figure 6B*, red square, VC-AC-hM4Di-CNO-VS-Baseline [6.6 ± 1.7%] vs. VC-AC-hM4Di-CNO-VS-Post intervention [6.2 ± 1.9%], 95% CI of difference [–15.0% to 15.9%], p>0.9999, n = 7, two-way RM ANOVA with post hoc Bonferroni test, see *Supplementary file 1* for detailed statistics). Same inactivation did not disrupt the association between AS and foot shock (*Figure 6B*, red circle, VC-AC-hM4Di-CNO-AS-Baseline [10.0 ± 2.3%] vs. VC-AC-hM4Di-CNO-AS-Post intervention [69.8 ± 5.1%], 95% CI of difference [44.4% to 75.2%], p<0.0001, n = 7; two-way RM ANOVA with post hoc Bonferroni test, see *Supplementary file 1* for detailed statistics). These results further proved that the VC-to-AC projection is also essential for the visuo-auditory association.

## The inactivation of the EC-to-AC CCK⁺ projection disrupts the behavioral recall of visuo-auditory associative memory

Given the crucial role of the EC-to-AC CCK⁺ projection in establishing visuo-auditory association, we further explored its involvement in memory recall within a behavioral context. We modified the above protocol. After VS and AS pairing in three consecutive days, we performed the baseline tests for the freezing response to both AS and VS on day 4. Subsequent to these baseline tests, the mice were fear-conditioned to the AS. On day 5, before testing the freezing responses to the AS and VS, we infused the CNO in the AC of the mice that had hM4Di expression in the CCK⁺ neurons of the EC (*Figure 6C*). We found that inactivation of the EC-to-AC CCK⁺ projection did not interfere with the fear memory response to the AS (*Figure 6D*, beige circle, EC-AC-hM4Di-CNO-AS-Baseline [13.6 ± 3.8%] vs. EC-AC-hM4Di-CNO-AS-Post intervention [58.2 ± 14.0%], 95% CI of difference [14.3% to 74.8%], p=0.0014, n = 6; two-way RM ANOVA with post hoc Bonferroni test, see *Supplementary file 1* for detailed statistics). However, this inactivation did disrupt the memory retrieval of associative memory between VS and AS (*Figure 6D*, beige square, EC-AC-hM4Di-CNO-VS-Baseline [12.0 ± 3.0%] vs. EC-AC-hM4Di-CNO-VS-Post intervention [14.5 ± 2.8%], 95% CI of difference [–32.7% to 27.7%], p>0.9999, n = 6, two-way RM ANOVA with post hoc Bonferroni test, see *Supplementary file 1* for detailed statistics).

In addition, we explored the role of the projection the VC-to-AC in the memory recall of the association between AS and VS. We employed a similar protocol to the previous experiment, but with a critical variation: on day 5, prior to testing the freezing responses to VS and AS, we specifically targeted the VC-to-AC projection that expressing hM4Di by bilaterally infusing CNO into the AC to silence this pathway (*Figure 6D*). Our results showed that inactivation of the VC-to-AC projection did not disrupt the fear memory response to AS (*Figure 6D*, red circle, VC-AC-hM4Di-CNO-AS-Baseline [13.3 ± 1.9%] vs. VC-AC-hM4Di-CNO-AS-Post intervention [68.5 ± 5.9%], 95% CI of difference [27.3% to 83.2%], p<0.0001, n = 7; two-way RM ANOVA with post hoc Bonferroni test, se *Supplementary file 1* for detailed statistics). Nonetheless, this inactivation significantly impaired the recall of associative memory between VS and AS (*Figure 6D*, red square, VC-AC-hM4Di-CNO-VS-Baseline [11.9 ± 4.0%] vs. VC-AC-hM4Di-CNO-VS-Post intervention [16.2 ± 4.0%], 95% CI of difference [–32.3% to 23.7%],

p=0.9986, n = 7, two-way RM ANOVA with post hoc Bonferroni test, see *Supplementary file 1* for detailed statistics).

These findings indicate that in addition to their role in establishing visuo-auditory associations, both the EC-to-AC CCK$^+$ projection and the VC-to-AC projection are crucial for retrieving recent associative memory.

## Application of a CCKBR antagonist blocks formation of the visuo-auditory association

The above results demonstrate that the formation of visuo-auditory associations in a behavioral context requires CCK$^+$ inputs from the EC. We have also demonstrated that HFS laser stimulation of EC CCK$^+$ neurons induced the release of endogenous CCK in the AC in *Figure 2—figure supplement 1T–W*. To further validate the role of CCK in the visuo-auditory memory task, we infused either L-365,260 or ACSF as a control into the AC. This was followed by a session of 25 pairings of VS with AS. Same as above, we repeated the above drug and pairing session four times per day and on three consecutive days. On day 4, baseline tests for the freezing response to the AS and VS were performed (three trials) before the mouse was fear-conditioned to the AS. After fear conditioning, freezing responses to the AS and VS were further examined on day 5 (*Figure 6—figure supplement 1A*). As expected, mice showed no freezing response to the AS before conditioning, but a high freezing rate to the AS after conditioning (*Figure 6—figure supplement 1B*, ACSF-AS-Baseline [7.4 ± 3.2%] vs. ACSF-AS-Post intervention [67.8 ± 4.0%], 95% CI of difference [47.4% to 73.3%], p<0.001, n = 9; L-365,260-AS-Baseline [5.8 ± 1.6%] vs. L-365,260-AS-Post intervention [67.5 ± 4.1%], 95% CI of difference [47.9% to 75.4%], p<0.001, n = 8; two-way RM ANOVA with post hoc Bonferroni test, see *Supplementary file 1* for detailed statistics). The ACSF mice group showed a significantly increased freezing response to the VS (*Figure 6—figure supplement 1B*, blue, ACSF-VS-Baseline [4.1 ± 0.9%] vs. ACSF-VS-Post intervention [24.4 ± 3.8%], 95% CI of difference [12.0% to 28.7%], p<0.001, n = 9, two-way RM ANOVA with post hoc Bonferroni test, see *Supplementary file 1* for detailed statistics). However, the bilateral infusion of L-365,260 into the AC blocked this association, resulting in a nil response to the VS (*Figure 6—figure supplement 1B*, red, L-365,260-VS-Baseline [4.3 ± 1.2%] vs. L-365,260-VS-Post intervention [2.5 ± 1.6%], 95% CI of difference [–7.1% to 10.6%], p=1.000, n = 8, two-way RM ANOVA with post hoc Bonferroni test, see *Supplementary file 1* for detailed statistics), which indicated that an association between the VS and AS was not established. There was a significant difference between the freezing rates to the VS of experimental and control groups (*Figure 6—figure supplement 1B*, L-365,260-VS-Post intervention [2.5 ± 1.6%, n = 8] vs. ACSF-VS-Post intervention [24.4 ± 3.8%, n = 9], 95% CI of difference [12.8% to 31.1%], p<0.001, two-way RM ANOVA with post hoc Bonferroni test, see *Supplementary file 1* for detailed statistics). These results demonstrate that the CCKBR antagonist prevented the generation of the association between the VS and AS and suggest an essential role of CCK in the generation of the visuo-auditory association.

## Systemic administration of CCK-4 rescues the deficit of *Cck$^{-/-}$* mice in visuo-auditory memory

As seen after treatment with a CCK antagonist, we expected that *Cck$^{-/-}$* mice would show a deficit in the formation of associative memory, and we tested this hypothesis (*Figure 6—figure supplement 1C*). Our previous results demonstrated that 6–9 trials were needed for *Cck$^{-/-}$* mice to produce a freezing rate of >60% in response to the conditioned AS, whereas only 3 trials were needed for wild-type mice, suggesting a general associative learning deficit in the *Cck$^{-/-}$* mice (*Chen et al., 2019*). The *Cck$^{-/-}$* mice in the control group (with saline injection) consistently showed a minimal freezing response to VS after visuo-auditory association and fear conditioning (*Figure 6—figure supplement 1D*, blue). To determine whether systemic administration of CCK could rescue this deficit, we administrated CCK-4 through a drug infusion cannula implanted into the transverse sinus. There is evidence that the tetrapeptide CCK-4 can penetrate the blood–brain barrier (*Rehfeld, 2000*). CCK-4 injection resulted in a significantly higher freezing rate compared to the controls (*Figure 6—figure supplement 1D*, red, CCK-VS-Post intervention [41.9 ± 5.6%, n = 7] vs. Saline-VS-Post intervention [10.0 ± 1.1%, n = 7], 95% CI of difference [19.5% to 44.2%], p<0.001; two-way RM ANOVA with post hoc Bonferroni test, see *Supplementary file 1* for detailed statistics), indicating that the visuo-auditory association was rescued upon CCK-4 administration.

To better compare the strength of visuo-auditory association under different experimental conditions, we calculated the ratio of the freezing response to the VS compared with that to the AS after conditioning (*Figure 6—figure supplement 1E*). The ratio of the CCKBR antagonist (L-365,260)-treated group was the lowest among all groups, demonstrating a nearly complete abolishment of the visuo-auditory association. Interestingly, the ratio of the CCK-4 group was the highest among all groups (*Figure 6—figure supplement 1E*, $F_{(3, 27)}$ = 28.797, CCK-4 infusion in CCK$^{-/-}$ mice [69.5 ± 6.2%, n = 7] vs. ACSF infusion in the wildtype mice [38.8 ± 7.3%, n = 9], 95% CI of difference [9.5% to 51.8%], p=0.002, one-way ANOVA with post hoc Bonferroni test, see *Supplementary file 1* for detailed statistics). This result indicates a possible compensatory upregulation of CCK receptors in *Cck$^{-/-}$* mice, leading to the highest association between the VS and AS, findings that are worth further investigation.

## Discussion

In the present study, we show that a direct input from the VC to the AC can be enhanced following pairing with postsynaptic firing triggered by auditory stimuli in the presence of CCK (*Figure 6E*). This potentiation was evident with both exogenous CCK administration and after applying HFS laser to the CCK$^+$ EC-to-AC projection, which resulted in the release of endogenous CCK. Notably, significant potentiation of the presynaptic input was achieved after just five pairing trials, even when the presynaptic activation occurred 200 ms prior to postsynaptic firing. Additionally, our behavioral experiments demonstrated that inactivating the EC-to-AC CCK$^+$ projection not only hindered the formation of visuo-auditory associations but also impaired the recall of these recent associative memories.

### Critical projections

Cross-modal association can be considered as the potentiation of synaptic strength between different modalities. Consistent with other studies (*Bizley et al., 2007*; *Budinger et al., 2006*; *Falchier et al., 2002*; *Falchier et al., 2010*; *Rockland and Ojima, 2003*), we describe a direct projection in the mouse from the VC to the AC for the visuo-auditory association using both retrograde and anterograde-tracing methods. The projection terminates both in the superficial and deep cortical layers. Our previous study on mouse demonstrated that neurons in the EC retrogradely labeled by true blue injected into the AC are almost 100% CCK$^+$ (*Li et al., 2014*). We here confirm that CCK is expressed in neurons of the EC and released to the AC.

### Cortical neuropeptides

Cortical neurons express a number of neuropeptides (*Somogyi and Klausberger, 2005*), whereby CCK is the most abundant of all. CCK comes in different forms, but it is the sulphated octapeptide, CCK-8S, that predominates in the brain (*Dockray et al., 1978*; *Rehfeld, 1978*). CCK is expressed in GABAergic interneurons (*Houser et al., 1983*), and many pyramidal neurons (*DeFelipe and Fariñas, 1992*) also have high levels of *Cck* transcript (*Burgunder and Young, 1988*; *Schiffmann and Vanderhaeghen, 1991*). The CCK$^+$ interneurons are relatively few, but exert a critical control of cortical activity (*Somogyi and Klausberger, 2005*). However, it is CCK in the pyramidal neurons that are in focus in the present study, especially the CCK$^+$, glutamatergic projection from EC-to-AC. In our experiments, we also use exogenous CCK in the experiments, both CCK-8S and CCK-4, which are at the C-terminus not only of proCCK but also of gastrin. The small size of the latter fragment is the reason why it is considered to pass the blood–brain barrier (*Rehfeld, 2000*). We infused CCK-4 into the transverse venous sinus aiming at obtaining maximal peptide levels in the cortex.

### Optogenetics

The present study is based on optogenetics, that is, genetic introduction of light-sensitive channels (Channelrodopsins), allowing control of selective neuron populations by light – a method that has revolutionized neuroscience research (*Deisseroth et al., 2006*; *Knöpfel et al., 2010*). Here, by using the two channels Chronos and ChrimsonR, we were able to activate two distinct projection terminals converging in the same target area (the EC-to-AC CCK$^+$ projection and the VC-to-AC projection).

### Prerequisites for synaptic plasticity

Our previous finding based on in vivo intracellular recording indicated that there are three prerequisites to enable synaptic plasticity: presynaptic activation, postsynaptic firing, and, in this particular

system, the presence of CCK (*Li et al., 2014*). Replacing the classical HFS protocol (HFS laser of the VC-to-AC projection) with local infusion of CCK-8S followed by pairing between laser stimulation-induced presynaptic VC-to-AC inputs and postsynaptic firing evoked by noise stimuli led to the potentiation of the VC-to-AC inputs. We hypothesize that these events may underlie the visuo-auditory association observed in the AC, further demonstrating the critical role of CCK to enable synaptic plasticity (*Chen et al., 2019*; *Li et al., 2014*).

Simple pairing between HFS laser of the EC-to-AC CCK[+] projection and presynaptic activation of the VC-to-AC projection without postsynaptic activation did not induce LTP in the VC-to-AC inputs. Neither did low-frequency (1 Hz) laser stimulation of the EC-to-AC CCK[+] projection induce LTP of the VC-to-AC inputs, probably since insufficient CCK was released by low-frequency stimulation. Surprisingly, no LTP was recorded after HFS laser of the VC-to-AC projection followed by Pre/Post Pairing. We demonstrate that the AC-projecting neurons in the VC have lower *Cck* expression compared to those in the EC. This could be a reason why LTP was not observed for the VC-to-AC inputs after only five trails of HFS laser VC-to-AC + Pre/Post Pairing. If the number of pairing trials or if laser stimulation intensity reaches a certain level, enough CCK may be released from the CCK[+] VC-to-AC-projecting neurons, and LTP may occur.

These findings suggest that in traditional LTP HFS also activates CCK[+] projection terminals, thereby releasing CCK and enabling potentiation (*Chen et al., 2019*). Subsequent experiments, in which the frequency of repetitive laser stimulation of the EC-to-AC CCK[+] projection terminals was changed, showed that the degree of potentiation increased with increasing frequency. This finding can be explained by the frequency-dependent nature of neuropeptide (CCK) release (*Bean and Roth, 1991*; *Hökfelt, 1991*; *Iverfeldt et al., 1989*; *Shakiryanova et al., 2005*; *Whim and Lloyd, 1989*).

In addition, the potentiation of the VC-to-AC inputs decreased as the interval between the termination of HFS laser and Pre/Post Pairing increased in the positive direction. A time window of 535 ms was observed to produce significant potentiation. If we would have increased the number of pairing trials, the effective Delay 1 might have lengthened.

## Hebbian plasticity

We addressed the issue of STDP and the critical time limit. Here the Hebbian rule has, arguably, been the most influential theory in learning and memory. This rule says that in order to induce potentiation the presynaptic subthreshold input should occur at most 20 ms before postsynaptic firing (*Bi and Poo, 1998*; *Markram et al., 1997*; *Zhang et al., 1998*). However, this theory has been challenged (*Drew and Abbott, 2006*; *Izhikevich, 2007*), with a prevalent question being: how can associations be established across behavioral time scales of seconds or even longer if the critical window is only 20 ms? (*Bittner et al., 2017*). Bittner et al. reported that five pairings of presynaptic subthreshold inputs with postsynaptic calcium plateau potentials produce a large potentiation, and that presynaptic inputs can arrive seconds before or after postsynaptic activity, a phenomenon termed 'behavioral time scale synaptic plasticity' (*Bittner et al., 2017*).

This may account for the highly plastic nature of place fields in the hippocampus. In agreement, we observed potentiation in the neocortex, even when presynaptic input arrived 200 ms (Delay 2) earlier than postsynaptic firing and after only five trials pairing, but only in the presence of CCK. This time window could perhaps be extended as the paring trials increase. In general terms, these results fit with the fact that neuropeptides are known to exert slow and long-lasting effects (*van den Pol, 2012*). We did not explore the reverse direction, in which postsynaptic activity occurred before presynaptic activity, because noise stimuli induced more than one spike and thus timing would be difficult to control.

## Limitations

In the behavioral part of our study, we focused exclusively on male subjects, which presents a notable limitation. Gender differences in brain function and response to stimuli are well-documented, suggesting that our findings might not fully extend to female subjects. This exclusion may limit the generalizability of our results. Future research including both genders would be valuable to understand more comprehensively the neurobiological mechanisms involved in visuo-auditory associations. Acknowledging this limitation, our study paves the way for subsequent research to explore potential gender-specific variations and their implications.

The natural condition that can activate the EC-to-AC projection similarly to the HFS laser is pending future exploration. We anticipate that stimuli with high valence, such as the air puff used in our prior studies (*Sun et al., 2022*; *Liang et al., 2023*), may significantly activate this pathway. Future studies can further examine how other areas are integrated to form cross-modality associative memory ultimately. In summary, we found that a direct projection from the VC to the AC provide an anatomical basis for visuo-auditory association. The VC-to-AC inputs was potentiated after pairing with postsynaptic firing evoked by the AS in the presence of CCK that was applied either exogenously or (endogenously) released from the EC-to-AC CCK+ projection terminals stimulated with an HFS laser. In the presence of endogenous CCK, significant potentiation of presynaptic input could be induced even if it arrived 200 ms earlier than postsynaptic firing and after only five trials of pairing. Finally, through the behavior experiments, we proved that the EC-to-AC CCK+ projection is important for both establishing and retrieving the recent visuo-auditory associative memories.

# Materials and methods

## Key resources table

| Reagent type (species) or resource | Designation | Source or reference | Identifiers | Additional information |
|---|---|---|---|---|
| Peptide, recombinant protein | Alexa Fluor 488-conjugated Cholera Toxin Subunit B | Thermo Fisher Scientific | Cat# C34775 | |
| Sequence-based reagent | Mm-Slc17a7-C2 | Advanced Cell Diagnostics | Cat# 416631-C2 | |
| Sequence-based reagent | Mm-Cck-C1 | Advanced Cell Diagnostics | Cat# 402271-C1 | |
| Sequence-based reagent | Mm-Tomato-C4 | Advanced Cell Diagnostics | Cat# 317041-C4 | |
| Recombinant DNA reagent | AAV9-Syn-ChrimsonR-tdTomato | UNC Vector Core | N/A | |
| Recombinant DNA reagent | AAV9-Ef1α-Flex-Chronos-GFP | UNC Vector Core | N/A | |
| Recombinant DNA reagent | AAV9-Syn-hM4Di-EGFP-WPRE-PA | Taitool BioScience | N/A | |
| Recombinant DNA reagent | AAV9-Syn-EGFP-WPRE-pA | Taitool BioScience | N/A | |
| Recombinant DNA reagent | AAVretro-hSyn-Cre-WPRE-hGH | A gift from James M. Wilson | Addgene viral prep #105553-AAVrg, RRID:Addgene_105553 | |
| Recombinant DNA reagent | AAV9-EF1a-DIO-EYFP-WPRE | BrainVTA | Cat# PT-0899 | |
| Recombinant DNA reagent | AAV9-Syn-DIO-ChrimsonR-mCherry-WPRE-Hgh | BrainVTA | Cat# PT-1374 | |
| Recombinant DNA reagent | AAV9-Syn-DIO-hM4Di-EYFP-WPRE-hGH pA | BrainVTA | Cat# PT-0043 | |
| Recombinant DNA reagent | AAV9-Syn-Cck2.3 | BrainVTA | Cat# PT-1629 | |
| Genetic reagent (*Mus musculus*) | Mouse: C57BL/6J | The Laboratory Animal Services Centre, Chinese University of Hong Kong; Laboratory Animal Research Unit, City University of Hong Kong | RRID:IMSR_JAX:000664 | |
| Genetic reagent (*M. musculus*) | Mouse: B6.Cg-Tg(Camk2a-cre)T29-1Stl/J | The Jackson Laboratory | RRID:IMSR_JAX:005359 | |
| Genetic reagent (*M. musculus*) | Mouse: B6.Cg-Gt(ROSA)26Sor$^{tm14(CAG-tdTomato)Hze}$/J (Ai14) | The Jackson Laboratory | RRID:IMSR_JAX:007914 | |
| Genetic reagent (*M. musculus*) | Mouse: *Cck*$^{tm1.1(cre)Zjh}$/J (*Cck*$^{Ires-Cre}$) | The Jackson Laboratory | RRID:IMSR_JAX:012706 | |

*Continued on next page*

*Continued*

| Reagent type (species) or resource | Designation | Source or reference | Identifiers | Additional information |
|---|---|---|---|---|
| Genetic reagent (*Rattus norvegicus*) | Rat: Sprague–Dawley | The Laboratory Animal Services Centre; Chinese University of Hong Kong | N/A | |
| Chemical compound, drug | Pentobarbital sodium (Dorminal 20%) | Alfasan International B.V. | N/A | |
| Chemical compound, drug | Urethane | Sigma-Aldrich | Cat# U2500 | |
| Chemical compound, drug | Lidocaine | Tokyo Chemical Industry | Cat# L0156 | |
| Chemical compound, drug | CNO | Sigma-Aldrich | Cat# C0832 | |
| Chemical compound, drug | CCK-4 | Abcam, Cambridge | Cat# ab141328 | |
| Chemical compound, drug | CCK-8S | Tocris Bioscience | Cat# 1166 | |
| Chemical compound, drug | L-365,260 | Tocris Bioscience | Cat# 2767 | |
| Software, algorithm | Fiji | https://imagej.net/software/fiji/ | RRID:SCR_002285 | |
| Software, algorithm | MATLAB R2020a | MathWorks | http://www.mathworks.com/products/matlab/; RRID:SCR_001622 | |
| Software, algorithm | ffline sorter | Plexon | http://www.plexon.com/products/offline-sorter; RRID:SCR_000012 | |
| Software, algorithm | Synapse suite | Tucker-Davis Technologies | https://www.tdt.com/component/synapse-software/ | |
| Software, algorithm | Origin | OriginLab | https://www.originlab.com/; | |
| Software, algorithm | SPSS | IBM | https://www.ibm.com/products/spss-statistics; RRID:SCR_019096 | |
| Other (stains) | Fluorescent Nissl Stain (Neurotrace 640) | Thermo Fisher Scientific | Cat# N21483; RRID:AB_2572212 | IHC 1:200 (To satin the Nissl bodies in the neurons, refer to section 'Histology') |
| Other | Guide Cannula | RWD Life Science | Cat# 62004 | Please refer to section 'Drug infusion' |
| Other | Dummy cannula (metal) | RWD Life Science | Cat# 62108 | Please refer to section 'Drug infusion' |
| Other | Internal injector | RWD Life Science | Cat# 62204 | Please refer to section 'Drug infusion' |
| Other | PE tube | RWD Life Science | Cat# 62329 | Please refer to section 'Drug infusion' |
| Other | Fiber Optic Cannula | Inper | Cat# FOC-W-L-6-20037 | Please refer to section 'Endogenous CCK release detection with fiber photometry' |

## Animals

In this study, we utilized adult C57BL/6J mice, CaMKIIa-Cre mice (Jackson lab stock #005359), Ai14 mice (Jackson lab stock #007914), *Cck* [Ires-Cre] mice (Jackson lab stock #012706), and Sprague–Dawley rats. For the behavioral experiments, we exclusively used male subjects. All animals were verified to have clean ears and normal hearing. They were kept under a controlled environment with a 12 hr-light/12 hr dark cycle, with the light period spanning from 8:00 pm to 8:00 am the following day. The temperature was maintained at 20–24°C and humidity levels were kept between 40 and 60%. All animals were provided with unlimited access to both food and water. All experimental procedures

received approval from the Animal Subjects Ethics Sub-Committee of the City University of Hong Kong (reference number of animal ethics review: A-59 and A-0467).

## Viruses

Adeno-associated viruses (AAVs) were purchased from the UNC Vector Core (AAV9-Syn-ChrimsonR-tdTomato, AAV9-Ef1α-Flex-Chronos-GFP), Taitool BioScience (AAV9-Syn-hM4Di-EGFP-WPRE-PA, AAV9-Syn-EGFP-WPRE-pA), Addgene (AAVretro-hSyn-Cre-WPRE-hGH [#105553-AAVrg]), and Brain VTA (AAV9-EF1a-DIO-EYFP-WPRE [PT-0899], AAV9-Syn-DIO-ChrimsonR-mCherry-WPRE-Hgh [PT-1374], AAV9-Syn-DIO-hM4Di-EYFP-WPRE-hGH pA [PT-0043], AAV9-Syn-Cck2.3 [PT-1629]).

## Surgery

For the procedure involving intracranial injection and the implantation of optical and drug cannulas, the animals were first prepared by administering an intraperitoneal (i.p.) injection of pentobarbital at a dosage of 80 mg/kg (Dorminal 20%, Alfasan International B.V., Woerden, the Netherlands). For the induction of anesthesia required for acute in vivo recordings, we utilized urethane at a concentration of 2 g/kg (i.p., Sigma-Aldrich, St. Louis, MO). Throughout the surgical procedure, we frequently administered 2% lidocaine (Tokyo Chemical Industry [TCI], #L0156, Tokyo, Japan) in droplet form directly onto the incision site to provide localized pain relief. To sustain anesthesia throughout the surgery, we periodically administered supplementary doses of either pentobarbital sodium or urethane. The animals were positioned on a stereotaxic instrument to ensure precision and stability. Prior to incision, the scalp area was thoroughly sterilized using 70% ethanol to prevent infection. A careful midline incision was then made, and the skull was carefully exposed. Meticulous adjustments were made to align the bregma and lambda reference points, as well as to ensure the leveling of the left and right sides. Following this preparation, craniotomies were then performed over the target brain regions, creating the necessary access for the subsequent procedure. Throughout the surgery, the animal's body temperature is maintained between 37 and 38°C using a heating blanket (Homeothermic Blanket System, Harvard Apparatus, USA) to prevent hypothermia. All surgical tools are sterilized via autoclaving before the procedure to uphold strict hygiene standards. Post-surgery, animals used for chronic experiments are closely monitored until they fully regain consciousness. Following recovery, they are returned to the Laboratory Animal Research Unit of the university for regular care. To prevent postoperative infections, erythromycin ointment is applied to the wound daily for a minimum of 1 wk.

## Brain microinjection of the AAVs or retrograde tracer

A pipette with a fine tip, loaded with AAV or retrograde tracer, is mounted on a Nanoliter 2000/Micro4 system (World Precision Instruments [WPI], Sarasota County, FL). To initiate the procedure, craniotomies of approximately 0.5–1 mm in diameter are performed over the targeted brain regions. The pipette is then carefully advanced to the target depth and held in place for 5 min to ensure stability before the infusion begins. The AAV or tracer is infused at a consistent rate of 25 nL/min, a standard speed for all injections to maintain precision. Following a 5 min infusion period, the pipette is gradually withdrawn, taking care to minimize potential damage to brain tissues.

## Drug infusion

For acute in vivo electrophysiological recordings, we utilized a 5 µL Hamilton syringe (#7105, Reno, NV) connected to a custom glass pipette via a flexible plastic tube (RWD Life Science #62329, Shenzhen, China). This assembly was operated by a syringe pump (KD Scientific #78-8210, Holliston, MA) to precisely infuse various drugs near the recording site in the AC. In chronic experiments, cannulas with an outer diameter of 0.41 mm (RWD Life Science #62004 and #62108, Shenzhen) were bilaterally implanted in the AC at specific coordinates: AP –2.9 mm, ML 0.35 mm medial to the juncture of the parietal and temporal skull bones, and DV –0.85 mm from the dura, on both the left and right sides. For bilateral drug infusion in the AC through these implanted cannulas, two 5 µL Hamilton syringes (#7105, Reno), connected to soft PE tubes (RWD Life Science #62329, Shenzhen) and corresponding injectors (RWD Life Science #62204, Shenzhen), were controlled by a dual syringe pump (KD Scientific #78-8210, Holliston) for the administration of different drugs. For drug infusion via the transverse sinus, we employed the same type of drug cannulas, implanting them directly above the sinus. The

infusion system utilized for this process mirrored that used for bilateral infusion in the AC through implanted cannulas.

## Retrograde tracing

For retrograde tracing, cholera toxin subunit B (CTB) with Alexa Fluor-488 conjugate (CTB488, 5 mg/mL, Fisher Scientific #C34775) was injected into the AC of rat (AP –3.5 mm [site 1], –4.5 mm [site 2], and –5.5 mm [site 3], ML 3.8 mm ventral to the edge differentiating the parietal and temporal skull, and DV –0.5 mm [depth 1] and –0.9 mm [depth 2], 50 nL for each of these six site-depth combinations). To quantify retrogradely labeled neurons, the Cell Counter plugin in Fiji was used. The cell density of a region was calculated by the total number of manually identified labeled neurons divided by the area.

## Quantification of CCK expression levels of AC-projecting neurons in the VC and the EC

To label the AC-projecting neurons in the VC and the EC, AAVretro-hSyn-Cre-WPRE-hGH (2.10E+13 vg/mL, Addgene, USA) was injected in three locations (100 nL/each) in the AC of Ai14 mice (AP –2.6 mm [site 1], –2.9 mm [site 2], and –3.2 mm [site 3], ML 1.0 mm ventral to the edge differentiating the parietal and temporal skull, and DV –0.5 mm from the dura). Three weeks later, the mice were deeply anesthetized with pentobarbital (Dorminal 20%, Alfasan International B.V.) and transcardially perfused with 20 mL of warm (37°C) 0.9% saline, 20 mL of warm fixative (4% paraformaldehyde, 0.4% picric acid, 0.1% glutaraldehyde in PBS), and 20 mL of the same ice-cold fixative. Brains were dissected out and postfixed in the same fixative for 24 hr at 4°C. The tissues were then washed three times with PBS and cryoprotected in 10% (overnight [O/N] at 4°C), 20% (O/N at 4°C), and 30% (O/N at 4°C) sucrose in PBS. Tissues were embedded in OCT compound, sectioned at 20 μm, and mounted onto Superfrost plus slides (Thermo Fisher Scientific, Waltham, MA). For in situ hybridization (RNAscope), the manufacturer's protocol was followed (Advanced Cell Diagnostics, San Francisco, CA). All experiments were replicated in three animals. The probes were designed by the manufacturer and available from Advanced Cell Diagnostics. The following probes were used in this study: Mm-Slc17a7- C2 (#416631-C2), Mm-Cck-C1 (#402271-C1), and Mm-Tomato-C4 (#317041-C4). All images were taken with the same settings on the same confocal microscope. The expression level of *Cck* in each animal was normalized against the mean *Cck* expression observed in the projecting neurons located within the VC. Neurons were categorized based on their *Cck* expression levels: those exhibiting *Cck* expression below the established average were defined as 'low *Cck*-expressing' neurons, whereas neurons with *Cck* expression surpassing this average value were classified as 'high *Cck*-expressing' neurons.

## in vivo fEPSP recording with optogenetics

To avoid photoelectric artifacts, we employed glass pipette electrodes (~1 MΩ) to record the fEPSPs when stimulating the projecting terminals that expressed various opsins in different experimental setups. In all our experimental setups, we followed a structured three-phase approach: initially, a baseline recording was conducted to establish the initial activity levels responding to different stimuli. This was followed by the manipulation phase, where specific experimental interventions were applied. Finally, recordings were made post-manipulation to observe and document the effects of the interventions. During both the baseline and post-manipulation recording phases of our experiments, the intensity of the stimuli was consistently maintained at the same level to ensure comparability. Additionally, a fixed ITI of 15 s was established for identical stimuli, allowing for sufficient recovery and response consistency. When using two or three types of stimuli, we interleaved them, spacing each evenly within the 15 s trial period for balanced presentation. Different intervention protocols were described as follows.

### Modified HFS protocol

We modified the HFS protocol and used four trials of 1 s laser pulse train (635 nm, the pulse width is 5 ms) at 80 Hz, each separated by a 10 se ITI, as illustrated in the upper part of *Figure 1D*. To activate the VC-to-AC projection, we injected AAV9-Syn-ChrimsonR-tdTomato (4.1 E+12 vg/mL, UNC Vector Core, Chapel Hill, NC) in two locations (150 nL/each) of the VC of wildtype mice. The coordinates were as follows: AP –2.7 mm (site 1) and –3.3 mm (site 2), ML 1.7 mm, DV –0.5 mm.

## Pre/Post Pairing protocol

The Pre/Post Pairing protocol entailed laser stimulation (635 nm, with a pulse width of 5 ms) of the projection from the VC to the AC to induce presynaptic activation, paired with a noise stimulus for postsynaptic firing. Given that the latency for the fEPSP from VC-to-AC is typically around 2–2.5 ms, and the latency for noise-induced responses in the AC of mice is generally 13 ms or more, we timed the laser stimulus to occur 10 ms following the noise stimulus. This timing ensured that presynaptic input was activated just prior to the onset of postsynaptic firing. We repeated this pairing sequence across a total of 80 trials.

## CCK (or ACSF) + Pre/Post Pairing protocol

Following the baseline recording, CCK-8S (10 ng/μL, 0.5 μL, 0.1 μL/min; Tocris Bioscience, Bristol, UK) or ACSF was infused into the AC. This was then followed by 80 trials of Pre/Post Pairing as described above. Subsequent to the completion of CCK (or ACSF)+ Pre/Post Pairing, we transitioned into the post-pairing recording phase.

## CCK alone, CCK + Pre, and CCK + Post protocols

The procedures for these protocols were consistent with the CCK + Pre/Post Pairing protocol, with these specific variations: (i) CCK alone: after the baseline recording, only CCK-8S was infused into the AC, without any subsequent Pre/Post Pairing; (ii) CCK + Pre: following the baseline recording, CCK-8S was infused into the AC. This was then followed by 80 trials of presynaptic activation, achieved through laser stimulation of the projection from the VC-to-AC; and (iii) CCK +Post: after the baseline recording and infusion of CCK-8S into the AC, the procedure was followed by 80 trials of noise stimuli.

## HFS laser VC-to-AC + Pre/Post Pairing protocol

To activate the VC-to-AC projection, we injected AAV9-Ef1α-Flex-Chronos-GFP (3.7 E+12 vg/mL, UNC Vector Core) in the VC (the coordinates and volume of virus were the same as above described) of CaMKIIα-Cre mice. The HFS laser VC-to-AC + Pre/Post Pairing protocol entailed applying HFS to the VC-to-AC projection, consisting of 10 pulses at 80 Hz, using 473 nm laser stimulation, where each pulse was 5 ms in width. After finishing the HFS, we waited for 10 ms before initiating the Pre/Post Pairing, as previously described. The presynaptic activation of the VC-to-AC projection was induced by 473 nm laser stimulation (with a pulse width of 5 ms). And the postsynaptic firing was evoked by noise stimulus. The HFS laser VC-to-AC + Pre/Post Pairing process was carried out for a total of five trials.

## HFS laser EC-to-AC + Pre/Post Pairing protocol

To activate the EC-to-AC CCK⁺ projection and the VC-to-AC projection, respectively, we injected the AAV9-Ef1α-Flex-Chronos-GFP in the EC (AP –4.2 mm, ML 3.5 mm and DV –3.0 mm, 300 nL), and AAV9-hSyn-ChrimsonR-tdTomato in the VC (the coordinates and volume of virus were the same as above described) of the $Cck^{Ires-Cre}$ mice. The HFS laser EC-to-AC + Pre/Post Pairing protocol entailed applying HFS to the EC-to-AC projection, consisting of 10 pulses at 80 Hz, using 473 nm laser stimulation, where each pulse was 5 ms in width. After finishing the HFS, we waited for 10 ms before initiating the Pre/Post Pairing, as previously described. The presynaptic activation of the VC-to-AC projection was induced by 635 nm laser stimulation (with a pulse width of 5 ms). And the postsynaptic firing was evoked by noise stimulus. The HFS laser EC-to-AC + Pre/Post Pairing process was carried out for a total of five trials.

## HFS laser EC-to-AC alone, HFS laser EC-to-AC + Pre, and HFS laser EC-to-AC + Post protocols

The protocols we employed were in line with the HFS laser EC-to-AC + Pre/Post Pairing protocol, with the following specific variations: (i) HFS laser EC-to-AC alone: after completing the baseline recording, we administered only the HFS laser targeting the EC-to-AC pathway. This was done without any subsequent Pre/Post Pairing and was carried out for a total of five trials. (ii) HFS laser EC-to-AC + Pre: following the baseline recording, HFS laser stimulation was applied to the EC-to-AC pathway. Then, 10 ms after the completion of the HFS, laser stimulation of the VC-to-AC projection was carried

out, but without the accompanying noise stimulus. The process was carried out for a total of five trials. (iii) HFS laser EC-to-AC+Post: post-baseline recording, we applied HFS laser to the EC-to-AC pathway. Subsequently, 10 ms following the termination of the HFS, a noise stimulus was introduced, but without the presynaptic activation of the VC-to-AC projection. The process was carried out for a total of five trials.

### Different variations of the HFS laser EC-to-AC + Pre/Post Pairing protocol

To investigate how varying parameters in this protocol influenced the potentiation of inputs from the VC to the AC, we focused on three specific variables: the frequency of HFS laser (referred to as 'frequency' with a default value of 80 Hz), the interval between the cessation of HFS laser stimulation and the onset of the Pre/Post Pairing (termed 'Delay 1' with a default value of 10 ms), and the time gap between presynaptic and postsynaptic activations (named 'Delay 2' with a default value of 0 ms). When adjusting any one of these variables, the other two were maintained at their respective default settings.

### L-365,260 (or ACSF) + HFS laser EC-to-AC + Pre/Post Pairing protocol

Following the baseline recording, L-365,260 (10 µg/mL, 0.5 µL, and 0.1 µL/min, Tocris Bioscience) or ACSF was infused into the AC. This was then followed by five trials of HFS laser EC-to-AC + Pre/Post Pairing as described above. After that, we transitioned into the post-pairing recording phase.

### High-Intensity Noise + Pre/Post Pairing protocol

In the High-Intensity Noise + Pre/Post Pairing protocol, we administered a 90 dB SPL noise stimulus and then waited for 10 ms prior to initiating the Pre/Post Pairing as above described. The presynaptic activation of the VC-to-AC projection was triggered using 635 nm laser stimulation (with a pulse width of 5 ms). Concurrently, the postsynaptic firing was evoked by the noise stimulus. This High-Intensity Noise + Pre/Post Pairing sequence was repeated for a total of five trials.

## Optimizing laser intensity for pathway-specific activation

To determine the optimal laser intensity for activating two distinct pathways, we conducted experiments on *Cck*[Ires-Cre] mice. In one set of experiments, we injected AAV9-Ef1α-Flex-Chronos-GFP into the EC, as depicted in *Figure 2—figure supplement 1A*. For another set, AAV9-Syn-ChrimsonR-tdTomato was injected into the VC, shown in *Figure 2—figure supplement 1B*. In both sets, we stimulated the terminals in the AC using 473 nm and 635 nm lasers at various intensity levels. In the EC-injected mice (*Figure 2—figure supplement 1A*, right), we observed that the fEPSP slopes progressively increased with the intensity of the 473 nm laser, reaching saturation at 30 mW/mm$^2$ (indicated by a green solid line). However, the 635 nm laser did not evoke any responses, even at an intensity as high as 40 mW/mm$^2$ (represented by a red dashed line). On the other hand, in VC-injected animals (*Figure 2—figure supplement 1B*, right), the fEPSP slopes increased and saturated with the 635 nm laser (red solid line). Interestingly, at 40 mW (green dashed line), the 473 nm laser could induce fEPSPs, but these responses were relatively small. Therefore, to minimize the likelihood of cross-activation between the two pathways, we carefully controlled the fiber end intensities of the 473 nm and 635 nm lasers, keeping them below 30 mW/mm$^2$ and 40 mW/mm$^2$, respectively.

## Brain slice preparation and patch-clamp recordings

At least 4 wk after virus injection, acute brain slices were prepared using a protective cutting and recovery method to achieve a higher success rate for patch clamp. Briefly, anesthetized mice received transcardial perfusion with NMDG-aCSF (92 mM NMDG, 2.5 mM KCl, 1.25 mM NaH$_2$PO$_4$, 30 mM NaHCO$_3$, 20 mM HEPES, 25 mM glucose, 2 mM thiourea, 5 mM Na-ascorbate, 3 mM Na-pyruvate, 0.5 mM CaCl$_2$·4H$_2$O, and 10 mM MgSO$_4$·7H$_2$O; pH 7.3–7.4), and the brain was gently extracted from the skull and then cut into 300-µm-thick sections. Slices were submerged in NMDG-aCSF for 5–10 min at 32–34°C to allow protective recovery and then incubated at room temperature ACSF (119 mM NaCl, 2.5 mM KCl, 1.25 mM NaH$_2$PO$_4$, 24 mM NaHCO$_3$, 12.5 mM glucose, 2 mM CaCl$_2$·4H$_2$O, and 2 mM MgSO$_4$·7H$_2$O, ~25°C) for at least 1 hr before transferring into recording chamber. All solutions were oxygenated with 95% O$_2$/5% CO$_2$ for 30 min in advance.

Whole-cell recordings were made from pyramidal neurons at the AC at room temperature. The signals were amplified with Multiclamp 700B amplifier, digitized with Digital 1440A digitizer, and acquired at 20 kHz using Clampex 10.3 (Molecular Devices, Sunnyvale, CA). Patch pipettes with a resistance between 3 and 5 MΩ were pulled from borosilicate glass (WPI) with a Sutter-87 puller (Sutter). The intracellular solution contained 145 mM K-gluconate, 10 mM HEPES, 1 mM EGTA, 2 mM MgATP, 0.3 mM Na2-GTP, and 2 mM $MgCl_2$; pH 7.3; 290–300 mOsm. The pipette was back-filled with internal solution containing 145 mM K-gluconate, 10 mM HEPES, 1 mM EGTA, 2 mM MgATP, 0.3 mM Na2-GTP, and 2 mM $MgCl_2$; pH 7.3; 290–300 mOsm.

Pyramidal neurons were identified based on their characteristic pyramidal-like morphology and a regular spiking firing pattern, determined by injecting a range of hyperpolarizing and depolarizing currents in 50 pA steps for 1 s each. An ES electrode was positioned approximately 200 μm from the recording neuron, and laser stimulations were administered via an optic fiber. Only those neurons responsive to both laser and ES were selected for subsequent experiments. In the voltage-clamp recording mode, with a holding potential of –70 mV, we recorded EPSCs elicited by both electrical (0.05 Hz, 0.5 ms) and laser (0.05 Hz, 5 ms) stimulations for a duration of 10 min. This was followed by HFS using a laser (80 Hz, five pulses, 5 ms pulse width) targeting either CCK[+] EC terminals (using 473 nm laser) or VC terminals (using 635 nm laser). After 10 ms, we paired presynaptic activation, induced by laser stimulation of the VC-to-AC projection, with postsynaptic activation triggered by ES of the AC. This pairing protocol was repeated five times at 10 s intervals. Subsequently, EPSCs induced by both electrical and laser stimulation were recorded for an additional 30 min. Throughout the recording, –5 mV hyperpolarizing pulses of 10 ms duration were applied every 20 s to monitor the access resistance (Ra). Recordings were terminated if Ra varied by more than 20%.

## Endogenous CCK release detection with fiber photometry

We administered an injection of AAV-Syn-DIO-ChrimsonR-mCherry (5.0 E+12 vg/mL, BrainVTA, Wuhan, China) into the EC (AP –4.2 mm, ML 3.5 mm, and DV –3.0 mm, 300 nL) of $Cck^{Ires-Cre}$ mice. Concurrently, AAV-Syn-Cck2.3 (5.2 E+12 vg/mL, BrainVTA) was injected into the AC of these mice. Following the virus injections, fiber optic cannulas (200 μm, 0.37 N/A, Inper, Hangzhou, China) were implanted in both the EC and AC, 10 min after virus injection. The tip of each cannula was positioned 100 μm above the site of virus infusion to ensure optimal light delivery. After a 3-week period, allowing for sufficient expression of the injected vectors, we proceeded to record the CCK dynamics (excitation light: 465 nm) and the isosbestic signal (excitation light: 405 nm) in the AC when activating the CCK[+] neurons in the EC with HFS laser stimulation (635 nm laser at 50 Hz).

## Visuo-auditory association task

VS lasting 5 s and AS lasting 3 s were initially paired and presented to mice. Each pair of stimuli was separated by a brief interval of 0.5 s. This pairing was repeated for 25 trials in each session, with an ITI of 30 s. Mice underwent four such sessions daily over a period of 3 d. On the fourth day, we conducted a baseline assessment, measuring the percentage of freezing behavior over a 10 s period following the separate presentations of VS and AS. Subsequently, the AS was paired with foot shocks, administered over three trials, to condition the mice. On the fifth day, post-conditioning tests were performed, evaluating freezing responses to both VS and AS independently. To assess the roles of different pathways and CCK in our experimental task, we conducted specific manipulations at various phases, as follows. (1) Inactivation of CCK[+] EC-to-AC projection during the encoding phase: In $Cck^{Ires-Cre}$ mice, we bilaterally injected AAV9-Syn-DIO-hM4Di-EYFP (300 nL, 5.70E+12 vg/mL, BrainVTA) or AAV9-EF1a-DIO-EYFP (300 nL, 5.24E+12 vg/mL, BrainVTA) into the EC (using the same coordinates as above), followed by bilateral implantation of drug cannulas in the AC. Three weeks later, behavioral tasks commenced. During the encoding phase, we inactivated the CCK[+] EC-to-AC projection by bilaterally infusing CNO (3 μM, Sigma-Aldrich #C0832, Darmstadt, Germany) into the AC before each day's VS and AS pairing for the first three days. (2) Inactivation of VC-to-AC projection during the encoding phase inactivation of VC-to-AC projection during the encoding phase: We bilaterally injected either AAV9-Syn-hM4Di-EGFP (4.5 E+12 vg/mL, Taitool BioScience, Shanghai, China) or AAV9-Syn-EGFP (1.42 E+13 vg/mL with a twofold dilution, Taitool BioScience) into the VC (the coordinates and volume of virus were the same as above described) of $Cck^{Ires-Cre}$ mice. After 3 wk, behavioral tasks began. To inactivate the VC-to-AC projection during encoding, CNO (3 μM) was infused into the AC prior to

the VS and AS pairing on the first three days. (3) Inactivation of CCK$^+$ EC-to-AC projection the during retrieval phase: Using a similar method as in step 1, we inactivated the CCK$^+$ EC-to-AC projection during the retrieval phase. CNO (3 µM) was infused into the AC before the freezing response tests to VS and AS on day 5. (4) Inactivation of VC-to-AC projection during the retrieval phase: Following the procedure in step 2, we inactivated the VC-to-AC projection during the retrieval phase. CNO (3 µM) was infused into the AC before day 5's freezing response tests to VS and AS. (5) Blockade of CCK signaling during the encoding phase in wildtype mice: Drug cannulas were bilaterally implanted in the AC of wildtype mice. After a 5-day recovery period, behavioral experiments commenced. To inhibit the CCK signaling pathway, L-365,260 (10 µg/mL, 0.5 µL at 0.1 µL/min, Tocris Bioscience) or ACSF was bilaterally infused into the AC before the pairing of visual and auditory stimuli (VS and AS) on the first three days. (6) Activation of CCK signaling in $Cck^{-/-}$ mice during the encoding phase: In $Cck^{-/-}$ mice, a drug infusion cannula was implanted atop the venous sinus (transverse sinus) to facilitate intravenous (i.v.) drug administration. Before the VS and AS pairing on the first three days, we injected either CCK-4 (0.01 mL, 3.4 µM) or saline (0.01 mL) through the implanted cannulas to activate the CCK signaling pathway.

## Histology

Animals were anesthetized with an overdose of pentobarbital sodium and subsequently perfused transcardially with 0.01 M phosphate-buffered saline (PBS) followed by 4% paraformaldehyde in 0.01 M PBS (4% PFA). The brains were extracted and post-fixed in 4% PFA for 24 hr. Brain tissues were sectioned at 60 µm thickness using a vibratome (Leica VT1000 S, Wetzlar, Germany). Nissl staining (Neurotrace 640, 1:200 in 0.01 M PBS with 0.1% Triton X-100, 2 hr, Thermo Fisher Scientific) was conducted to indicate the neurons. Imaging was performed using a Nikon Eclipse Ni-E upright fluorescence microscope (Tokyo, Japan) or a Zeiss LSM880 confocal microscope (Oberkochen, Germany).

## Acoustic stimuli, visual stimuli, electrical stimulation, and laser stimulation

In our experiments, all stimuli were generated using RZ6 and RZ5D processing stations (Tucker-Davis Technologies [TDT], FL) with control facilitated by Synapse software from the TDT Synapse suite. The acoustic stimuli, created as analog signals, were transmitted through a free-field magnetic speaker (MF-1, TDT). We calibrated the sound pressure level of these acoustic stimuli using a condenser microphone (Center Technology, Taipei). Visual stimuli, also in the form of analog signals, were presented via a white LED array. The ESs aimed at activating the AC were generated by the TDT system and administered through a low-impedance stimulation electrode, coupled with an isolator (ISO-Flex, A.M.P.I., Jerusalem, Israel). For laser stimulation, we utilized a laser generator (Inper, #B1465635), which was activated by signals sent from the Synapse-controlled RZ5D processor. The laser light was then directed to the specific target areas via a fiber patch cable (200 µm, 0.37 NA, Inper). Throughout these procedures, all events were captured and recorded by the TDT system, ensuring accurate documentation and analysis.

## Quantification of the electrophysiological data

The fEPSP slope (in vivo) and EPSC amplitude (ex vivo) were used to compare the response level before and after different manipulation protocols. To calculate the fEPSP slope (*Figure 1C*, upper), we initially identified the most linear segment (marked by a starting point t1 and an end point t2, typically ranging from 20% and 80% of the maximum amplitude) within the descending phase of the fEPSP. We then conducted a linear regression analysis using the data from this specific segment (spanning from t2 to t1). The absolute value of the slope of the resultant regression line was thus determined to be the slope of the fEPSP. For comparative analyses, baseline values of fEPSP slopes or EPSC amplitudes were first averaged. Subsequent values were then normalized against this baseline.

## Statistical analysis

All statistical analyses were performed using SPSS software (IBM, USA). Pairwise comparisons were adjusted by Bonferroni correction. Statistical significance was set at $p<0.05$. Refer to *Supplementary file 1* for detailed statistics.

## Acknowledgements

This work was supported by Hong Kong Research Grants Council, General Research Fund: 11103220M, 11101521M (GRF, JFH); Hong Kong Research Grants Council, Collaborative Research Fund: C1043-21GF(CRF, JFH); Innovation and Technology Fund: MRP/053/18X, GHP_075_19GD (ITF, JFH); Health and Medical Research Fund: 06172456, 09203656 (HMRF, XC, JFH); The Swedish Research Council: 2018-0273 (TH); The Arvid Carlsson Foundation (TH); The Swedish Research Council: 220-01688 (TH); Key-Area Research and Development Program of Guangdong Province: 2018B030340001 (WJS); and the following charitable foundations for their generous support to JFH: Wong Chun Hong Endowed Chair Professorship, Charlie Lee Charitable Foundation, and Fong Shu Fook Tong Foundation.

## Additional information

### Funding

| Funder | Grant reference number | Author |
|---|---|---|
| Hong Kong Research Grants Council | General Research Fund 11103220M | Jufang He |
| Hong Kong Research Grants Council | General Research Fund 11101521M | Jufang He |
| Hong Kong Research Grants Council | Collaborative Research Fund C1043-21GF | Jufang He |
| Hong Kong Innovation and Technology Commission | Innovation and Technology Fund MRP/053/18X | Jufang He |
| Hong Kong Innovation and Technology Commission | Innovation and Technology Fund GHP_075_19GD | Jufang He |
| Hong Kong Health Bureau | Health and Medical Research Fund 06172456 | Xi Chen |
| Hong Kong Health Bureau | Health and Medical Research Fund 09203656 | Jufang He |
| Swedish Research Council | 2018-0273 | Tomas Hökfelt |
| Arvid Carlsson Foundation | | Tomas Hökfelt |
| Swedish Research Council | 220-01688 | Tomas Hökfelt |
| Key Area Research and Development Program of Guangdong Province | 2018B030340001 | Wenjian Sun |
| Wong Chun Hong Endowed Chair Professorship | | Jufang He |
| Charlie Lee Charitable Foundation | | Jufang He |
| Fong Shu Fook Tong Foundation | | Jufang He |

The funders had no role in study design, data collection and interpretation, or the decision to submit the work for publication.

### Author contributions

Wenjian Sun, Conceptualization, Data curation, Formal analysis, Funding acquisition, Investigation, Visualization, Methodology, Writing – original draft; Haohao Wu, Data curation, Formal analysis, Investigation, Visualization, Methodology; Yujie Peng, Formal analysis, Investigation, Visualization, Methodology; Xuejiao Zheng, Jing Li, Dingxuan Zeng, Formal analysis, Investigation; Peng Tang, Ming Zhao, Hemin Feng, Hao Li, Ye Liang, Investigation; Junfeng Su, Conceptualization; Xi Chen, Conceptualization, Resources, Software, Supervision, Funding acquisition, Validation, Investigation,

Methodology, Writing - review and editing; Tomas Hökfelt, Conceptualization, Resources, Supervision, Funding acquisition, Validation, Project administration, Writing - review and editing; Jufang He, Conceptualization, Resources, Data curation, Supervision, Funding acquisition, Validation, Project administration, Writing - review and editing

## Author ORCIDs
Wenjian Sun ![ORCID] https://orcid.org/0000-0003-1584-8242
Jing Li ![ORCID] http://orcid.org/0000-0001-8075-0650
Xi Chen ![ORCID] http://orcid.org/0000-0002-2144-6584
Jufang He ![ORCID] http://orcid.org/0000-0002-4288-5957

## Ethics
All experimental procedures were approved by the Animal Subjects Ethics Sub-Committees of City University of Hong Kong (Reference number of animal ethics review: A-59 and A-0467).

## Decision letter and Author response
Decision letter https://doi.org/10.7554/eLife.83356.sa1
Author response https://doi.org/10.7554/eLife.83356.sa2

---

# Additional files

## Supplementary files
- MDAR checklist
- Supplementary file 1. Detailed statistics related to all figures and figure supplements.

## Data availability
Source data files have been provided for Figures 1–6 and their figure supplements.

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
