## [Editor Report]

This fundamental work advances our understanding of how neuropeptides influence cortical circuits and cortical plasticity. The evidence supporting the conclusions is compelling. This work would be of interest to neuroscientists working on cortical processing and plasticity and general roles neuropeptides play in brain function.

---

## [Decision Letter]

**Decision letter after peer review:**

Thank you for submitting your article "Heterosynaptic Plasticity of the Visuo-auditory Projection Requires Cholecystokinin released from Entorhinal Cortex Afferents" for consideration by *eLife*. Your article has been reviewed by 2 peer reviewers, and the evaluation has been overseen by a Reviewing Editor and Kate Wassum as the Senior Editor. The reviewers have opted to remain anonymous.

Essential revisions:

In this manuscript, He and colleagues demonstrate the role of cholecystokinin (CCK)-positive neurons in the entorhino-auditory projection and the peptide CCK itself in the plasticity of visuo-auditory (VC-AC) association. Using a series of in vivo and ex vivo electrophysiology recordings the authors provide evidence that high-frequency stimulation of the CCK-positive neurons paired with an auditory stimulus can evoke long-term potentiation (LTP) of the visuo-auditory projection and temporal parameters of the plasticity were systematically explored. The observed plasticity also has behavioral relevance. The reviewers agree upon the significance and broad impact of the current work but also raised a substantial issue with control experiments summarized below:

1. Address whether CCK alone or CCK in a particular context would be sufficient for potentiation. This includes control experiments requested by Reviewer #1, points 1-4, and general comments by Reviewer #2.

2. The manuscript will need significant improvement in scientific communication to have better readability, method and figure description, and accuracy as suggested and detailed by both Reviewers.

3. Please provide a key resource table if you have not already.

4. Comment on the limitation of the exclusion of females from the behavioral study may benefit the manuscript.

With your revision, please provide a point x point response to each reviewer comment from both the public review and recommendations for authors.

*Reviewer #1 (Recommendations for the authors):*

1. To remove the ambiguity raised in the first comment in public review, in figure 1G the synaptic strength after the application of CCK but before pairing VALS and noise should be used as the baseline; in Figures 2E, 3G, control conditions of HFLSEA alone, HFLSEA+ VALS (Suppl. Figure 2D), and HFLSEA+noise/electrical stimulation should be added.

2. To deal with the issue raised in the third comment, the CCK receptor blocker can be added to the experiments in Figure 2E and 3G to see if the plasticity will be affected.

3. For comment 4, to examine the contribution of VC-AC and EC-AC projections in this visuo-auditory memory task (Figure 6), the authors can try disrupting the EC-AC or VC-AC projections during visuoauditory pairing and disrupting VC-AC projection during the final test.

4. The introduction was poorly written and hard to follow. The rationale, objective, and significance of the study were not well elaborated. I would like to recommend the authors refine it.

5. I couldn't find how the slope of fEPSP is calculated. Please illustrate it in Figure 1C.

6. In Figure 4A, the area highlighted by the box is mostly higher visual areas. The primary visual cortex is more lateral to it and ignored. Is there any reason behind this selection? For example, are retrogradely labeled neurons denser in higher visual areas than the primary visual area? Please quantify the density of retrogradely labeled neurons across areas and layers.

7. How are the axons coming from VC or EC distributed across layers? Related to the anatomy, from which layers were the recordings done? Does the plasticity of VC-AC projection measured by fEPSP (Figure 1,2) or EPSC (Figure 3) depend on layers? Please add details in the figures and text.

8. Typo: page 13, line 261, 'underly'.

*Reviewer #2 (Recommendations for the authors):*

Specific comments:

Figure 1

– 1D: Can ChrimsonR follow 80 Hz reliably? According to Klapoetke et al., 2014 fidelity already decreases to 75% at 60 Hz. Please insert appropriate citation or show in an experiment that ChrimsonR can follow.

– 1D: The baseline of the After condition is very unstable in the representative trace.

– 1G: There's no description in the methods of how CCK is infused. If it's infused immediately before pairing, an additional control of CCK without pairing is needed.

– 1G: After repeated exposure is there a rise of CCK release?

Figure 2

– 2C, D: There's no proper description in methods section about the sequence of the Chrimson and Chronos stimulation in relation to the VALS induction. Presumably, there is timed optogenetic stimulation to evoke EPSCs and the exact sequence and protocol is not described properly.

– It's not entirely clear whether AS has the same effect as HFLSVA?

– 2 E: It's not clear whether VALS is just a simple 5 sec stimulation; in which case it's surprising that HFLS of EC to AC alone can evoke LTP.

– 2F: the representative trace doesn't seem to show a 20% change: "Before" doesn't seem to be aligned, looks like "Before" is positioned a bit higher than "After".

– Do the same AC cells receive inputs from VC and EC or does CCK diffuse away from the release site?

Figure 3

– 3A: Because this experiment is performed in slice and not in in vitro cultures it should be referred to as ex vivo not in vitro.

– 3E: Why is 60 Hz being used instead of 80 Hz? It would be better to keep it consistent, as the same mechanism applies in vivo and ex vivo.

– 3I, L: There is no distribution of data points in the Before condition (no error bars) unlike other experiments e.g. Figure 2.

– 3I: Seems like significance is driven by one outlier.

Figure 4

– Does intensity correlate with expression level? Please cite relevant papers or use Q-PCR to verify correlation.

Figure 5

– The authors claim that HFLSEA causes CCK release which is essential for visuo-auditory association. However, the use of CCKBR antagonist in this experiment does not directly test the hypothesis that CCK is released from neurons originating from the EC nor that HFLS of the EC to AC projection is required.

– The y-axis is often cut off which makes it difficult to compare data from different graphs. It should start at 0 instead of arbitrary values.

– Heavy use of author-invented abbreviations makes it difficult to follow the paper. It would be better to use conventional abbreviations such as EC to AC and VC to AC for entorhino-auditory, visuo-auditory projection and e.g. HFS(laser) EC to AC instead of HFLSEA. A similar rationale applies to additional abbreviations like VALS and ESAC.

– A schematic of how the authors think CCK and auditory stimulation could work together on a molecular level would be helpful.

– In order to support the conclusion that CCK is released after HFLS, it is essential to show that CCK release is triggered by HFSLEA.

– Do the authors have a hypothesis about what natural stimulus (e.g. noise) HFLS in the entorhino-auditory projection and subsequent CCK release mimics?

– It would strengthen the authors' conclusion if they showed results of optogenetic inhibition of CKK^+^ neurons in the entorhino-auditory projection during the visuo-auditory association task.

– Additional analysis to strengthen the claim of the RNAscope experiment would be to quantify the number of cells which express CCK in the VC vs EC.

[Editors' note: further revisions were suggested prior to acceptance, as described below.]

Thank you for resubmitting your work entitled "Heterosynaptic Plasticity of the Visuo-auditory Projection Requires Cholecystokinin released from Entorhinal Cortex Afferents" for further consideration by *eLife*. Your revised article has been evaluated by Kate Wassum (Senior Editor) and a Reviewing Editor.

The manuscript has been improved but there are some remaining issues that need to be addressed, as outlined below:

- Please include full statistical reporting in the main manuscript including exact p-values wherever possible alongside the summary statistics (test statistic and df) and 95% confidence intervals. These should be reported for all key questions and not only when the p-value is less than 0.05.

---

## [Author Response]

Essential revisions:In this manuscript, He and colleagues demonstrate the role of cholecystokinin (CCK)-positive neurons in the entorhino-auditory projection and the peptide CCK itself in the plasticity of visuo-auditory (VC-AC) association. Using a series of in vivo and ex vivo electrophysiology recordings the authors provide evidence that high-frequency stimulation of the CCK-positive neurons paired with an auditory stimulus can evoke long-term potentiation (LTP) of the visuo-auditory projection and temporal parameters of the plasticity were systematically explored. The observed plasticity also has behavioral relevance. The reviewers agree upon the significance and broad impact of the current work but also raised a substantial issue with control experiments summarized below:

We really appreciate the reviewers’ constructive comments, which are very helpful for us to improve our study.

1. Address whether CCK alone or CCK in a particular context would be sufficient for potentiation. This includes control experiments requested by Reviewer #1, points 1-4, and general comments by Reviewer #2.

We have added the related control experiments to consolidate our conclusion. Please see the details in the responses to reviewers’ corresponding comments.

2. The manuscript will need significant improvement in scientific communication to have better readability, method and figure description, and accuracy as suggested and detailed by both Reviewers.

In response to the valuable suggestions, we have meticulously revised our manuscript, focusing particularly on the introduction, methods, and figure legends. Our aim was to enhance the accuracy and clarity of our descriptions. We hope that these revisions have significantly improved the overall readability of the manuscript.

3. Please provide a key resource table if you have not already.

A key resource table has been included in the manuscript, as requested, to provide comprehensive information on the resources used in our study.

4. Comment on the limitation of the exclusion of females from the behavioral study may benefit the manuscript.

We acknowledge the limitation arising from the exclusion of female subjects in our behavioral study. We understand that including both genders could provide a more comprehensive understanding of the visuo-auditory association and its neural underpinnings. This aspect will be considered in future research to ensure a more inclusive and representative analysis. We included a discussion of this limitation in our manuscript to provide a complete perspective on our study's scope and implications as follow.

“Gender limitation. In the behavioral part of our study, we focused exclusively on male subjects, which presents a notable limitation. Gender differences in brain function and response to stimuli are well-documented, suggesting that our findings might not fully extend to female subjects. This exclusion may limit the generalizability of our results. Future research including both genders would be valuable to understand more comprehensively the neurobiological mechanisms involved in visuo-auditory associations. Acknowledging this limitation, our study paves the way for subsequent research to explore potential gender-specific variations and their implications.**”**

With your revision, please provide a point x point response to each reviewer comment from both the public review and recommendations for authors.Reviewer #1 (Recommendations for the authors):1. To remove the ambiguity raised in the first comment in public review, in figure 1G the synaptic strength after the application of CCK but before pairing VALS and noise should be used as the baseline; in Figures 2E, 3G, control conditions of HFLSEA alone, HFLSEA+ VALS (Suppl. Figure 2D), and HFLSEA+noise/electrical stimulation should be added.

Please see the response to comment 1.

2. To deal with the issue raised in the third comment, the CCK receptor blocker can be added to the experiments in Figure 2E and 3G to see if the plasticity will be affected.

Please see the response to comment 3.

3. For comment 4, to examine the contribution of VC-AC and EC-AC projections in this visuo-auditory memory task (Figure 6), the authors can try disrupting the EC-AC or VC-AC projections during visuoauditory pairing and disrupting VC-AC projection during the final test.

Please see the response to comment 4.

4. The introduction was poorly written and hard to follow. The rationale, objective, and significance of the study were not well elaborated. I would like to recommend the authors refine it.

We have refined the introduction carefully and tried to make the rationale, objective, and significance clearer. Hope it is better now.

5. I couldn't find how the slope of fEPSP is calculated. Please illustrate it in Figure 1C.

In Figure 1C, we have depicted the method for measuring the slope of the fEPSP. To calculate this slope, we initially identify the most linear segment (marked by a starting point t1 and an end point t2, typically ranging from 20% and 80% of the maximum amplitude) within the descending phase of the fEPSP. We then conduct a linear regression analysis using the data from this specific segment (spanning from t2 to t1). The slope of the resultant regression line is thus determined to be the slope of the fEPSP.

6. In Figure 4A, the area highlighted by the box is mostly higher visual areas. The primary visual cortex is more lateral to it and ignored. Is there any reason behind this selection? For example, are retrogradely labeled neurons denser in higher visual areas than the primary visual area? Please quantify the density of retrogradely labeled neurons across areas and layers.

Thank you for your valuable feedback. In response, we have quantified the density of retrogradely labeled neurons across different areas and layers, as shown in Figure 1—figure supplement 1A. Our findings reveal a higher density of retrogradely labeled neurons in the higher order associative visual cortex (VC) compared to the primary VC. Notably, layer V emerged as the most predominant layer in terms of neuron density.

7. How are the axons coming from VC or EC distributed across layers? Related to the anatomy, from which layers were the recordings done? Does the plasticity of VC-AC projection measured by fEPSP (Figure 1,2) or EPSC (Figure 3) depend on layers? Please add details in the figures and text.

Thanks for your constructive comments and suggestions. The axons coming from VC and EC are distributed in both superficial and deep layers. We recorded across layer II/III to layer V (as indicated in Figure 1B). And we also added the following description in the main text: “The recording of fEPSPs was conducted across layers II/III to layer V within the auditory cortex (AC) (Figure 1B).”. The criterion for fEPSP recording is that the laser stimulation of VC-AC projection and EC-AC projection can both evoke responses in the recording site, as well as noise induced response. And the criteria of slice recording are (1) there are EPSCs of the patched cell to both laser stimulation of the VC-AC projection and electrical stimulation of AC; (2) in cases where no EPSC response was observed in the patched cell following laser stimulation of the EC-AC projection, it was crucial that other nearby cells within the same slice demonstrated EPSC responses to the stimulation.

Regarding the inquiry about whether the plasticity of the VC-to-AC projection is layer-dependent, our current dataset does not provide enough evidence for a definitive conclusion. However, this is an intriguing question and merits further investigation in future studies. We truly value this insightful suggestion and are appreciative of the opportunity to consider this aspect in our ongoing research.

8. Typo: page 13, line 261, 'underly'.

Thanks for pointing out that. We have corrected it.

Reviewer #2 (Recommendations for the authors):Specific comments:Figure 1– 1D: Can ChrimsonR follow 80 Hz reliably? According to Klapoetke et al., 2014 fidelity already decreases to 75% at 60 Hz. Please insert appropriate citation or show in an experiment that ChrimsonR can follow.

Thank you for highlighting this important concern. In response, we have conducted an evaluation of ChrimsonR's fidelity at 80 Hz. Our data demonstrate that stimulation with a 635 nm laser at 80 Hz, featuring a pulse width of 5 ms, reliably evokes fEPSPs in the AC when stimulating the VC-to-AC projection that expresses ChrimsonR, as depicted in Figure 1—figure supplement 1D. This observed difference in fidelity compared to the findings in the Klapoetke et al., 2014 study could be attributed to the distinct experimental setups. While the previous study focused on single-cell recordings on the in vitro cultured cells, our analysis was conducted using in vivo fEPSP recordings.

– 1D: The baseline of the After condition is very unstable in the representative trace.

Thank you for highlighting this issue. We acknowledge that the baseline in the 'After' condition of the representative trace in Figure 1D appeared unstable. In response to your observation, we have now replaced it with another set of representative traces that exhibit a stable baseline.

– 1G: There's no description in the methods of how CCK is infused. If it's infused immediately before pairing, an additional control of CCK without pairing is needed.

We have incorporated detailed information about the CCK infusion process in the method. Specifically, CCK was administered immediately prior to the pairing procedure. To ensure comprehensive analysis, we also included a control experiment where CCK was infused without subsequent pairing. This additional experiment and its results are presented in Figure 1—figure supplement 1F-G for reference.

– 1G: After repeated exposure is there a rise of CCK release?

In the experiment demonstrated in Figure 1G, since we infused CCK-8S in the AC to see the effect of CCK in the potentiation of VC-AC projection we perhaps can’t measure if CCK is released or not in a clear way.

To test the release of CCK, we injected AAV-syn-CCKsensor (AAV-syn-Cck2.3) in the AC and AAV9-syn-Flex-ChrimsonR-mcherry in the EC of *Cck*
^Ires-Cre^ mice, and implanted optic fiber in the AC and EC to record the CCK release in the AC by fiber photometry when stimulate the EC CCK^+^ neurons. Our results demonstrated that HFS laser of the EC CCK^+^ neurons induced a consistent CCK signal in the AC, indicating that endogenous CCK is released in the AC in response to HFS laser stimulation of the CCK^+^ neurons in the EC (Figure 2—figure supplement 1T-W).

Figure 2– 2C, D: There's no proper description in methods section about the sequence of the Chrimson and Chronos stimulation in relation to the VALS induction. Presumably, there is timed optogenetic stimulation to evoke EPSCs and the exact sequence and protocol is not described properly.

Thank you for pointing out this issue. We have added more detailed descriptions about different protocols in the methods. For comprehensive information on these protocols, please refer to the expanded details now included in the methods.

“HFS laser VC-to-AC + Pre/Post Pairing

To activate the VC-to-AC projection, we injected AAV9-Ef1α-Flex-Chronos-GFP into the VC of CaMKIIα-Cre mice. After conducting baseline recordings, we proceeded with the HFS laser VC-to-AC + Pre/Post Pairing protocol. This protocol entailed applying HFS to the VC-to-AC projection, consisting of 10 pulses at 80 Hz, using 473 nm laser stimulation, where each pulse was 5 ms in width. After finishing the HFS, we waited for 10 ms before initiating the Pre/Post Pairing, as previously described. The presynaptic activation of the VC-to-AC projection was induced by 473 nm laser stimulation (with a pulse width of 5ms). And the postsynaptic firing was evoked by noise stimulus. The HFS laser VC-to-AC + Pre/Post Pairing process was carried out for a total of 5 trials. Subsequently, we proceeded with the post-pairing recording phase.

HFS laser EC-to-AC + Pre/Post Pairing

To activate the EC-to-AC CCK^+^ projection and the VC-to-AC projection respectively, we injected the AAV9-Ef1α-Flex-Chronos-GFP in the EC, and AAV9-hSyn-ChrimsonR-tdTomato in the VC of the *Cck*
^Ires-Cre^ mice. After conducting baseline recordings, we proceeded with the HFS laser EC-to-AC + Pre/Post Pairing protocol. This protocol entailed applying HFS to the EC-to-AC projection, consisting of 10 pulses at 80 Hz, using 473 nm laser stimulation, where each pulse was 5 ms in width. After finishing the HFS, we waited for 10 ms before initiating the Pre/Post Pairing, as previously described. The presynaptic activation of the VC-to-AC projection was induced by 635 nm laser stimulation (with a pulse width of 5ms). And the postsynaptic firing was evoked by noise stimulus. The HFS laser EC-to-AC + Pre/Post Pairing process was carried out for a total of 5 trials. Subsequently, we proceeded with the post-pairing recording phase.”

– It's not entirely clear whether AS has the same effect as HFLSVA?

Thank you for your valuable feedback. In response, we have incorporated a set of control experiments designed to evaluate whether AS yield the same effect as high-frequency laser stimulation of the visual cortex to auditory cortex. This involved presenting high-intensity noise (90dB) prior to the Pre/Post pairing, as documented in Figure 2—figure supplement 1D-G. Our findings suggest that such generalized high-intensity stimuli do not result in the potentiation of the VC-to-AC projection, nor do they enhance the noise response.

– 2 E: It's not clear whether VALS is just a simple 5 sec stimulation; in which case it's surprising that HFLS of EC to AC alone can evoke LTP.

We apologize that we didn’t make it clear. Laser stimulation to the VC-to-AC projection is 5 ms laser stimulation to the terminals of VC-AC projection. In Figure 2E, the red color represented the responses to laser stimulation of the VC-to-AC projection before and after “HFS laser EC-to-AC + Pre/Post Pairing” as shown in Figure 2C instead of “HFS laser EC-to-AC” alone. We have corrected it in revised version.

– 2F: the representative trace doesn't seem to show a 20% change: "Before" doesn't seem to be aligned, looks like "Before" is positioned a bit higher than "After".

Thanks for highlighting the issue. We have replaced it with a better pair of representative traces**.**

– Do the same AC cells receive inputs from VC and EC or does CCK diffuse away from the release site?

Based on our slice recording, 7 out of 14 patched AC cells simultaneously receive inputs from both VC and EC. Since CCK is a neuropeptide, it is probably released extrasynaptically and could diffuse away from the release site. And the VC-AC inputs were still potentiated for those cells without direct EC inputs after HFS laser EC-to-AC + Pre/Post Pairing.

Figure 3– 3A: Because this experiment is performed in slice and not in in vitro cultures it should be referred to as ex vivo not in vitro.

Thanks for pointing out this mistake. We have corrected it in the revised manuscript.

– 3E: Why is 60 Hz being used instead of 80 Hz? It would be better to keep it consistent, as the same mechanism applies in vivo and ex vivo.

Thank you for pointing out this inconsistency. Following your observation, we have conducted a repeat of the experiment using an 80Hz frequency to align with our in vivo experiments. The conclusions drawn from this 80 Hz experiment are consistent with those obtained from the initial 60 Hz experiment. For more detailed information, please refer to Figure 3.

– 3I, L: There is no distribution of data points in the Before condition (no error bars) unlike other experiments e.g. Figure 2.

Thank you for pointing out the discrepancy. We have revised our statistical approach for the slice recording data to ensure consistency with the methods used in our in vivo experiments. This change has been implemented in the updated version of the manuscript. Please see the Figure 3I and 3L attached above.

– 3I: Seems like significance is driven by one outlier.

Thank you for highlighting this concern. In response to your observation regarding the 60 Hz experiment, we identified and removed the data point with the maximum value and reanalyzed the results using a two-way repeated measure ANOVA with Bonferroni post hoc test. This reanalysis yielded a p-value of 0.025 for the pairwise comparison between HFLSEA-Before and HFLSEA-After, affirming its statistical significance. Additionally, the p-value for the pairwise comparison between HFLSEA-After and HFLSVA-After was 0.039, also indicating significance.

In the updated version of our manuscript, as previously mentioned, we have substituted the 60 Hz experiment with a new 80 Hz experiment, effectively addressing the issue of potential outlier-driven significance.

Figure 4– Does intensity correlate with expression level? Please cite relevant papers or use Q-PCR to verify correlation.

Thank you for addressing this important query. The RNAscope assay, which we employed for quantitative RNA detection, is well-established in determining gene expression levels. Its sensitivity, specificity, and accuracy have been extensively validated in various studies, including those by Caldwell et al., 2021; Jolly et al., 2019; and Chan et al., 2018. We have duly incorporated these references into the revised manuscript to substantiate the reliability of the RNAscope assay in correlating intensity with expression level.

Caldwell, C., Rottman, J.B., Paces, W. et al. Validation of a DKK1 RNAscope chromogenic in situ hybridization assay for gastric and gastroesophageal junction adenocarcinoma tumors. Sci Rep 11, 9920 (2021). https://doi.org/10.1038/s41598-021-89060-3

Jolly, S., Lang, V., Koelzer, V.H. et al. Single-Cell Quantification of mRNA Expression in The Human Brain. Sci Rep 9, 12353 (2019). https://doi.org/10.1038/s41598-019-48787-w

Chan S, Filézac de L’Etang A, Rangell L, Caplazi P, Lowe JB, Romeo V (2018) A method for manual and automated multiplex RNAscope in situ hybridization and immunocytochemistry on cytospin samples. PLoS ONE 13(11): e0207619. https://doi.org/10.1371/journal.pone.0207619

Figure 5– The authors claim that HFLSEA causes CCK release which is essential for visuo-auditory association. However, the use of CCKBR antagonist in this experiment does not directly test the hypothesis that CCK is released from neurons originating from the EC nor that HFLS of the EC to AC projection is required.

Thanks for pointing out the issue. In response, to build the link that HFS laser of the EC-to-AC CCK^+^ projection is important for visuo-auditory association in the behavioral context, we conducted the following additional behavioral studies (for details please see the response to comment 4 of reviewer 1):

1) Assessing the Necessity of CCK^+^ EC-to-AC Projection in Establishing Visuo-Auditory Associative memories, by inactivating the pathway with inhibitory DREADD during the encoding phase.

2) Investigating the Importance of CCK^+^ EC-to-AC Projection in Recalling Visuo-Auditory Association, by inactivating the pathway with inhibitory DREADD during the retrieving phase.

For CCK release, we injected AAV-syn-CCKsensor in the AC and AAV9-syn-Flex-ChrimsonR-mcherry in the EC of *Cck*
^Ires-Cre^ mice, and implanted optic fiber in the AC and EC to record the CCK release in the AC by fiber photometry when stimulate the EC CCK^+^ neurons. Our results demonstrated that HFS laser of the EC CCK^+^ neurons induced a consistent CCK signal in the AC, indicating that endogenous CCK is released in the AC in response to HFS laser stimulation of the CCK^+^ neurons in the EC (Figure 2—figure supplement 1T-W).

– The y-axis is often cut off which makes it difficult to compare data from different graphs. It should start at 0 instead of arbitrary values.

Thank you for highlighting this issue. We have revised our approach to the presentation of bar plots. Now, in all bar plots, we have ensured that the y-axis starts at 0 for clarity and consistency, with the exception of Figure 5. Due to the multitude of parameters involved in Figure 5, we have started the y-axis at 80% and extended it to 160% to more clearly illustrate the differences. Additionally, within each figure, we have maintained consistent y-axis scales across all plots for ease of comparison.

– Heavy use of author-invented abbreviations makes it difficult to follow the paper. It would be better to use conventional abbreviations such as EC to AC and VC to AC for entorhino-auditory, visuo-auditory projection and e.g. HFS(laser) EC to AC instead of HFLSEA. A similar rationale applies to additional abbreviations like VALS and ESAC.

Thank you for bringing this issue to our attention. In the updated manuscript, we have replaced the author-invented abbreviations with more conventional and widely recognized terms. This change is intended to enhance the clarity and readability of the paper.

– A schematic of how the authors think CCK and auditory stimulation could work together on a molecular level would be helpful.

Thanks for this constructive feedback. We have made a schematic to show how the endogenous CCK work with the auditory stimulation to enable the potentiation of the VC AC projection (Figure 6E).

– In order to support the conclusion that CCK is released after HFLS, it is essential to show that CCK release is triggered by HFSLEA.

As noted earlier, we have utilized a CCK sensor combined with fiber photometry to measure CCK release, with the findings detailed in Figure 2—figure supplement 1T-W. For more comprehensive information on this methodology, please refer to our response to comment 4 (in related to Figure 1G) from reviewer 2.

– Do the authors have a hypothesis about what natural stimulus (e.g. noise) HFLS in the entorhino-auditory projection and subsequent CCK release mimics?

At present, our hypothesis is that in an awake state, stimuli that are novel, intense, and carry significant positive or negative valence could be analogous to the HFS of the EC-to-AC projection and the consequent release of CCK. We have included this hypothesis and a discussion of its implications in a new, simplified paragraph in the revised manuscript.

– It would strengthen the authors' conclusion if they showed results of optogenetic inhibition of CKK^+^ neurons in the entorhino-auditory projection during the visuo-auditory association task.

Thanks for pointing out the issue. In response, to build the link that HFS laser of the EC-to-AC CCK^+^ projection is important for visuo-auditory association in the behavioral context, we conducted the following additional behavioral studies (for details please see the response to comment 4 of reviewer 1):

1) Assessing the Necessity of CCK^+^ EC-to-AC Projection in Establishing Visuo-Auditory Associative memories, by inactivating the pathway with inhibitory DREADD during the encoding phase.

2) Investigating the Importance of CCK^+^ EC-to-AC Projection in Recalling Visuo-Auditory Association, by inactivating the pathway with inhibitory DREADD during the retrieving phase.

– Additional analysis to strengthen the claim of the RNAscope experiment would be to quantify the number of cells which express CCK in the VC vs EC.

Thank you for your constructive suggestion. In response, we have quantified the number of AC-projecting neurons expressing *Cck* in both the VC and EC. This additional data has been incorporated into the main text to strengthen the findings from our RNAscope experiment.

“In terms of cell count, we identified 345 *CcK^+^* AC-projecting neurons in the EC and 199 in the VC across three animals. Out of these, 217 out of 345 cells in the EC and 81 out of 199 cells in the VC showed elevated *Cck* transcripts.”

[Editors’ note: what follows is the authors’ response to the second round of review.]

The manuscript has been improved but there are some remaining issues that need to be addressed, as outlined below:- Please include full statistical reporting in the main manuscript including exact p-values wherever possible alongside the summary statistics (test statistic and df) and 95% confidence intervals. These should be reported for all key questions and not only when the p-value is less than 0.05.

We have included full statistical reporting in the main manuscript. For example:

*“*I Al, 2). Many retrogradely labeled neurons were observed in both the primary and associative VC (Figure IA3, 4), but more were observed in the associative than the primary VC and were mainly distributed in the layer V (Figures IA4 and Figure I—figure supplement IA, two-way ANOVA, F (4, 40) = 4.707, significant interaction, p = 0.0033, n 5; F (1:40) = 9.768, primary [6.3±1.0] vs. associative [11.1±2.2], p 0.0033, 95% confidence interval [CI] of difference, [7.8 to -1.7]; F – 17.18, different layers, p < 0.0001; Bonferroni's multiple comparison test, Layer I-Associative VC [0.7±0.4] vs. Layer V-Associative VC [28.9±4.7], p<0.0001, 95% CI of difference, [-40.2 to -16.2]; Layer 11/111-Associative VC [6.3±1.7] vs. Layer V-Associative VC [28.9±4.7], p<0.0001, 95% CI of difference, [-34.5 to -10.6]; Layer IV-Associative VC [10.7±2.9] vs. Layer V-Associative VC [28.9±4.7], p 0.0002, 95% CI of difference, [-30.2 to 6.3]; Layer VI-Associative VC [8.8±2.1] vs. Layer V-Associative VC [28.9±4.7], p<0.0001, 95% CI of difference, [-32.1 to -8.2], see Supplementary File I for detailed statistics. The result here…”